# Inhibition of ACSS2-mediated histone crotonylation alleviates kidney fibrosis via IL-1β-dependent macrophage activation and tubular cell senescence

Lingzhi Li[1,4], Ting Xiang[1,4], Jingjing Guo[2], Fan Guo[1], Yiting Wu[1], Han Feng[3], Jing Liu[1], Sibei Tao[1], Ping Fu [1]✉ & Liang Ma [1]✉

Histone lysine crotonylation (Kcr), as a posttranslational modification, is widespread as acetylation (Kac); however, its roles are largely unknown in kidney fibrosis. In this study, we report that histone Kcr of tubular epithelial cells is abnormally elevated in fibrotic kidneys. By screening these crotonylated/acetylated factors, a crotonyl-CoA-producing enzyme ACSS2 (acyl-CoA synthetase short chain family member 2) is found to remarkably increase histone 3 lysine 9 crotonylation (H3K9cr) level without influencing H3K9ac in kidneys and tubular epithelial cells. The integrated analysis of ChIP-seq and RNA-seq of fibrotic kidneys reveal that the hub proinflammatory cytokine IL-1β, which is regulated by H3K9cr, play crucial roles in fibrogenesis. Furthermore, genetic and pharmacologic inhibition of ACSS2 both suppress H3K9cr-mediated IL-1β expression, which thereby alleviate IL-1β-dependent macrophage activation and tubular cell senescence to delay renal fibrosis. Collectively, our findings uncover that H3K9cr exerts a critical, previously unrecognized role in kidney fibrosis, where ACSS2 represents an attractive drug target to slow fibrotic kidney disease progression.

Chronic kidney disease (CKD) is a condition characterized by functional deterioration with sustained inflammation, and progressive fibrosis of the kidneys, affecting over 800 million people worldwide[1,2]. As known, macrophage activation is a common feature of inflammation in active fibrotic kidney lesions. Accelerated tubular cell senescence via the release of components of senescence-associated secretory phenotypes (SASPs) also promotes the pathogenesis of kidney fibrosis[3]. Since current treatments to slow CKD progression are limited and nonspecific, it eventually progresses to end-stage kidney disease, which requires dialysis or kidney transplantation[4]. Therefore, exploring the precise mechanism of CKD to find potential drug targets is of great importance to ultimately arrest and prevent its progression.

Protein posttranslational modifications (PTMs) play essential roles in modifying protein function and thus influence numerous biological processes, including organismal development, cell differentiation, cell death, and inflammation[5]. The aberrant PTMs in fibrosis and inflammation are becoming more crucial in the progression of CKD[6], and histone lysine acetylation (Kac) is one of the most well-studied PTMs. With the development of mass spectrometry (MS) techniques[7,8], lysine crotonylation (Kcr) was explored to be a evolutionarily conserved PTM principally related to active

[1]Department of Nephrology, Institute of Kidney Diseases, West China Hospital of Sichuan University, and National Key Laboratory of Kidney Diseases, Chengdu, China. [2]Department of Urology, Institute of Urology, West China Hospital of Sichuan University, Chengdu, China. [3]Tulane Research and Innovation for Arrhythmia Discoveries-TRIAD Center, Tulane University School of Medicine, New Orleans, LA, USA. [4]These authors contributed equally: Lingzhi Li, Ting Xiang. ✉e-mail: fupinghx@scu.edu.cn; liang_m@scu.edu.cn

transcription, where histones are the most abundant crotonylated proteins[9]. Histone Kcr, specifically enriched at promoters and potential enhancers in the mammalian genome, exhibits a stronger effect on transcription than histone Kac[10]; however, Kcr and Kac are catalyzed by several similar enzymes and metabolic states, which makes it difficult to distinguish them and explore the role of crotonylation alone.

In the context of kidney diseases, few studies reported that histone Kcr was observed in both healthy and diseased kidneys, including acute kidney injury (AKI)[9] and IgA nephropathy[11], suggesting that it may play key roles in epigenetic regulation of gene expression in kidney diseases. Recently, nuclear condensation of a chromodomain Y-like transcription corepressor (CDYL) has linked histone Kcr to transcriptional responses and cystogenesis in autosomal dominant polycystic kidney disease (ADPKD). However, there is little knowledge regarding the functional role of histone Kcr relative to Kac during kidney injury and fibrosis.

In this study, we aimed to explore the function of histone crotonylation-especially histone 3 lysine 9 crotonylation (H3K9cr)-in patients and mice with kidney fibrosis as well as tubular epithelial cells (TECs). Our results reveal that H3K9cr is involved in kidney fibrosis by regulating macrophage activation and senescence of TECs. We propose that the crotonyl-CoA-producing enzyme, acyl-CoA synthetase short-chain family member 2 (ACSS2), could selectively regulate H3K9cr level and function, which represents an attractive drug target for strategies to slow fibrotic kidney disease progression.

## Results

### The level of histone Kcr positively correlates with the severity of CKD in patients and mice

To explore the roles of protein crotonylation in human kidneys, we first collected kidney biopsy slides determined the clinical characteristics of CKD patients, and performed immunohistochemistry (IHC) using antibodies targeting crotonyl lysine (anti-Kcr) (Fig. 1A and

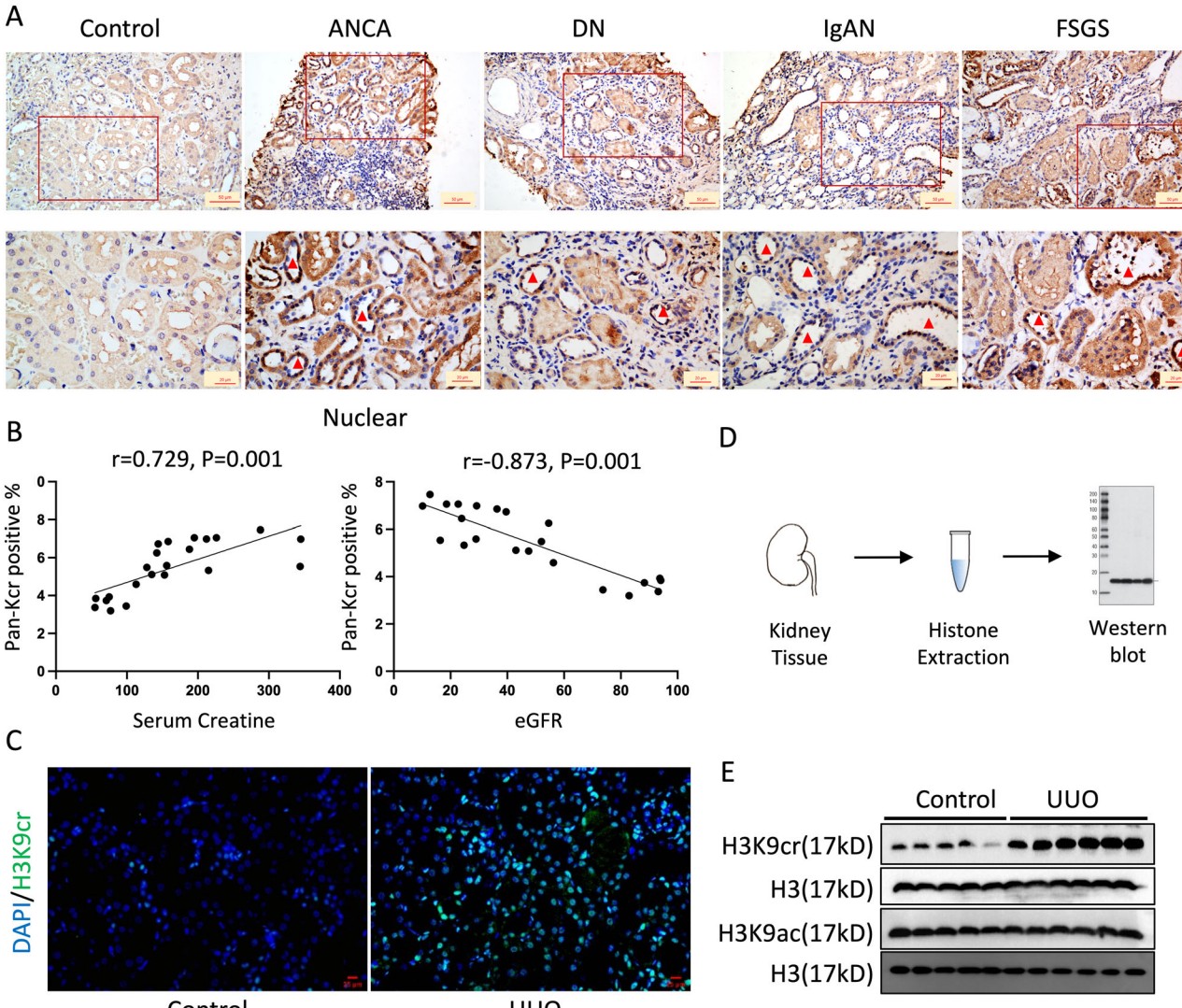

**Fig. 1 | The increased levels of crotonylation in human renal biopsies of CKD and fibrotic kidneys from mice. A** Representative IHC staining of pan-Kcr antibody in healthy control subjects and patients with chronic kidney diseases. Scale bar: upper panels: 50 μm; lower panels:20 μm. **B** Quantitative IHC analysis of pan-Kcr level in renal nuclear from control and patients with chronic kidney diseases using ImageJ 6.0 software; using Pearson correlation statistical analysis, two-sided statistical tests. (n = 6 for ANCA, n = 9 for DN, n = 3 for IgA, n = 3 for FSGS). **C** IF staining of H3K9cr (green) and DAPI (blue) in control and UUO kidneys. (n = 3 per group). Scale bar: 20 μm. **D** Flow chart of the process to collect histone. **E** Protein expression of H3K9cr, H3K9ac and H3 in fibrotic kidneys of UUO mice were determined by western blotting (n = 5 to 6 animals per group). IHC immunohistochemical, pan-Kcr pan anti-crotonyllysine, IF Immunofluorescence, FAN folic acid nephropathy, UUO unilateral ureteric obstruction, ANCA anti-neutrophil cytoplasmic antibodies, DN diabetic nephropathy, IgAN IgA nephropathy, FSGS focal segmental glomerulosclerosis. Triangle: representative positive staining of pan-Kcr. Statistical analysis by Pearson correlation.

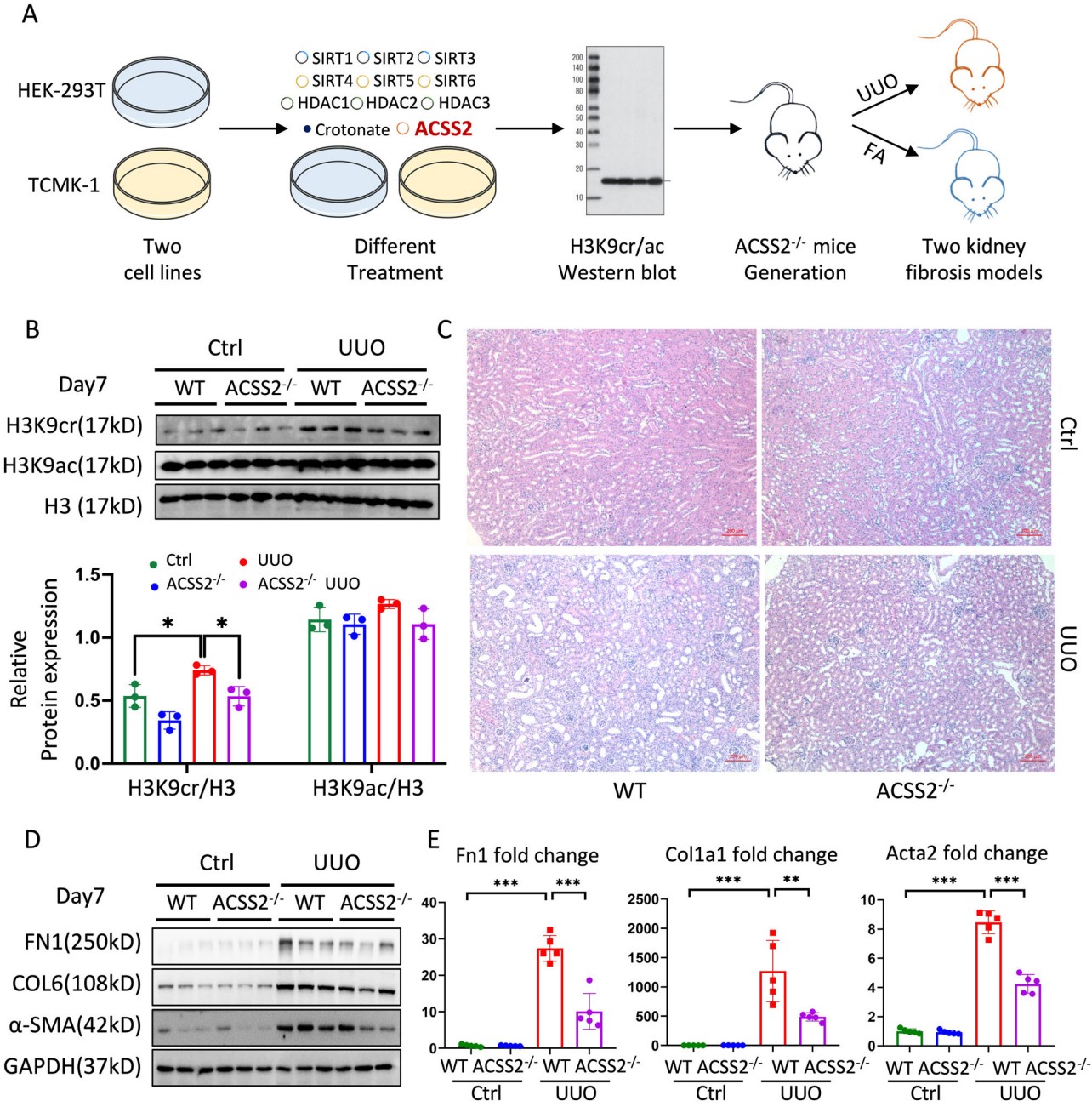

**Fig. 2 | Global deletion of ACSS2 suppressed H3K9cr level and alleviated kidney fibrosis in mice. A** Schematic overview of process for screening factors influencing H3K9cr in vitro and experimental design in ACSS2$^{-/-}$ mouse. **B** Protein level and quantitative analysis of H3K9cr, H3K9ac, and H3 in whole kidney lysates of control and UUO of WT and ACSS2$^{-/-}$ mice ($n = 3$ per group, Control $vs.$ UUO: $p = 0.032$; UUO $vs.$ ACSS2KO + UUO: $p = 0.03$). **C** Representative images of H&E staining in UUO of WT and ACSS2$^{-/-}$ mice ($n = 6$ per group). Scale bar: 200 µm. **D** FN1, COL6, α-SMA, and GAPDH immunoblotting in the whole kidney lysates of control and UUO of WT

and ACSS2$^{-/-}$ mice ($n = 3$ per group). **E** mRNA levels of Fn1, Col1a1 and Acta2 in whole kidney lysates of WT and ACSS2$^{-/-}$ mice treated with UUO ($n = 5$ per group, Control $vs.$ UUO: $p < 0.001$ for Fn1, $p < 0.001$ for Col1a, $p < 0.001$ for Acta2; UUO $vs.$ ACSS2$^{-/-}$ + UUO: $p < 0.001$ for Fn1, $p = 0.007$ for Col1a, $p < 0.001$ for Acta2). FAN: folic acid nephropathy; UUO: unilateral ureteric obstruction; WT: wild type; ACSS2$^{-/-}$: ACSS2 knockout; Ctrl: Control. Data shown are means ± SEM. Statistical analysis by one-way ANOVA with Tukey's post hoc test. *$p < 0.05$, **$p < 0.01$ and ***$p < 0.001$.

Supplementary Fig. 1A). The basic characteristics of the patients are summarized in Supplementary Data 1. The results exhibited that Kcr was mainly present in TECs, and was significantly higher in kidneys of CKD patients than that of control (Fig. 1A). The IHC results showed that the intensity of Kcr was positively correlated with disease progression, especially samples with nuclear staining (Fig. 1B and Supplementary Fig. 1B). Even when adjusting age and sex, the intensity of nuclear Kcr was positively correlated with serum creatine and negatively correlated with eGFR (Supplementary Data 2). The Kcr was involved in CKD patients as well as contributed to fibrosis in the obstructed kidneys of

mice. The immunofluorescence (IF) staining data exhibited that unilateral ureteric obstruction (UUO) induced the increasing level of Kcr, which was mainly colocalized in the nuclei (Supplementary Fig. 1C). To confirm the changes in histone Kcr, we extracted total histones from the fibrotic kidneys of mice and evaluated the histone purification using Coomassie brilliant blue staining and liquid chromatography-mass spectrometry (LC-MS/MS) analysis. The LC-MS/MS analysis confirmed that all the proteins were extracted from the nucleus and functioned with chromatin structure and dynamics (Supplementary Fig. 2A–C). To compare the levels of histone crotonylation in kidneys,

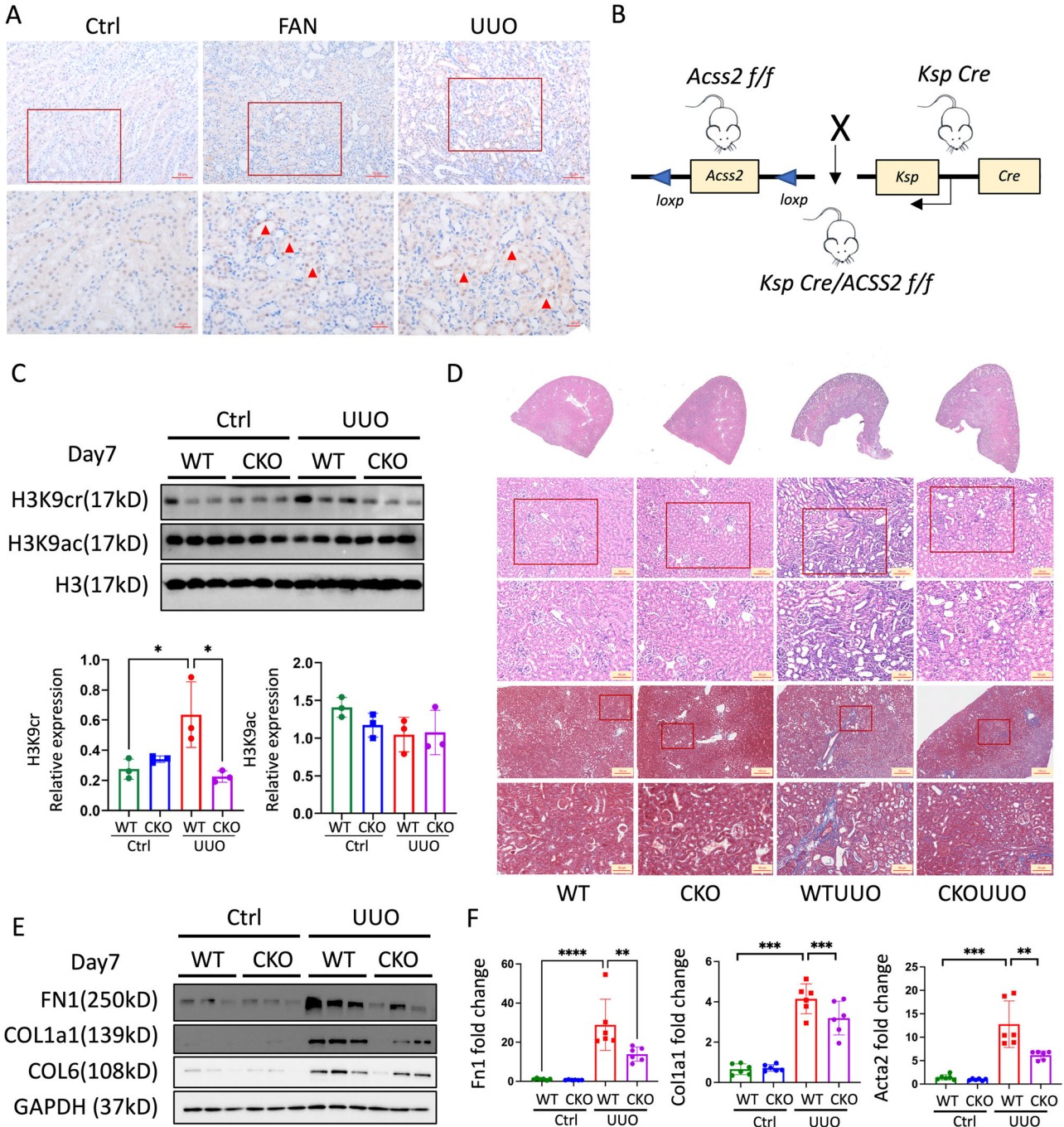

**Fig. 3 | TEC-specific deletion of ACSS2 suppressed H3K9cr level and alleviated fibrosis in mice. A** Representative IHC staining of ACSS2 in kidneys of control and mice treated with UUO or injected with FAN (*n* = 3 per group). Scale bar: upper panels: 50 μm; lower panels: 20 μm. **B** Design strategy of TEC-specific deletion of ACSS2 (ACSS2[tecKO]) mice. **C** H3K9cr, H3K9ac and H3 immunoblotting and quantification of these immunoblots in the whole kidney lysates of control and UUO of WT and ACSS2[tecKO] mice (*n* = 3 per group, Control vs UUO: *p* = 0.0212; UUO vs ACSS2CKO + UUO: *p* = 0.0106). **D** Representative images of H&E and Masson staining in kidneys of UUO of WT and ACSS2[tecKO] mice (*n* = 3 per group). Scale bar: upper panels: 100 μm; lower panels: 50 μm. **E** FN1, COL1a1, COL6 and GAPDH

immunoblotting in the whole kidney lysates of control and UUO of WT and ACSS2[tecKO] mice (*n* = 3 per group). **F** mRNA levels of Fn1, Col1a1, Acta2 or Col6 in whole kidney lysates of WT and ACSS2[tecKO] mice treated with UUO (*n* = 6 per group, Control *vs.* UUO: *p* < 0.0001 for Fn1, *p* < 0.0001 for Col1a, *p* < 0.0001 for Acta2; UUO *vs.* ACSS2CKO + UUO: *p* = 0.0055 for Fn1, *p* = 0.0011 for Col1a, *p* = 0.0456 for Acta2). UUO unilateral ureteric obstruction, WT wild type, IHC Immunohistochemical, ACSS2 CKO tubular epithelial cell-specific deletion of ACSS2. Triangle: representative positive staining of ACSS2. Data shown are means ± SEM. Statistical analysis by one-way ANOVA with Tukey's post hoc test. **p* < 0.05, ***p* < 0.01, ****p* < 0.001 and *****p* < 0.0001.

western blotting analysis using anti-Kcr antibodies was performed (Fig. 1D and Supplementary Fig. 1D, E), demonstrating that the increase of histone crotonylation was accompanied with kidney fibrosis in both UUO and folic acid nephropathy (FAN) mice (Supplementary Fig. 2D–G).

As previously reported, different histone lysine modifications might exert diverse roles. To identify histone Kcr and Kac at several histone H3 residues reported (K9, K14, K18, and K27), we used different antibodies to blot the histone extraction. In particular, the level of H3K9cr increased remarkably in two experimental mice models of

kidney fibrosis, while H3K9ac remained stable level (Fig. 1E, Supplementary Fig. 1F, and Supplementary Fig. 1H, I), proposing that H3K9cr may have distinct effects relative to H3K9ac in renal fibrosis. In addition, the level of H3K9cr was also increased in human fibrotic kidney slides and negatively correlated with eGFR in humans (Supplementary Fig. 1G). In addition, H3K18cr and H3K27cr, corresponding with H3K18ac and H3K27ac, both increased in fibrotic kidneys, suggesting that these two markers are coregulated. Interestingly, H3K14cr increased in kidneys with FAN and UUO; H3K14ac showed opposite trends, recommending that the roles of histone Kcr and Kac are both complicated and comprehensive (Supplementary Fig. 2H–K). However, we mainly focused on the level and function of H3K9cr in further study of kidney fibrosis.

### Genetic deletion of ACSS2 decreased the level of H3K9 crotonylation and alleviated kidney fibrosis

As histone Kcr and Kac are reversible, dynamic processes mediated by multiple identical enzymes, it has been challenging to distinguish their individual functions. Based on the abovementioned data, the alterations of H3K9cr appear to be independent of those of H3K9ac in kidney fibrosis. We examined these known enzymatic and metabolic regulation variables, including Sirtuin (SIRT) 1/2/3/4/5/6[9,12], histone deacetylase (HDAC) 1/2/3[13,14], acyl-CoA synthetase short chain family member 2 (ACSS2)[15] and crotonate in vitro, to identify critical factors that could regulate H3K9cr more than H3K9ac (Fig. 2A) in both mouse renal tubular epithelial (TCMK-1) cells, and human embryonic kidney (HEK-293T) cells. By screening these factors, ACSS2 overexpression by plasmid transfection (Supplementary Fig. 3A, B) dramatically increased H3K9cr level and did not change H3K9ac, indicating that ACSS2 may exert a greater impact on H3K9cr than H3K9ac (Supplementary Fig. 4B).

Crotonate, a source of histone Kcr, also increased the level of H3K9cr and H3K9ac in TECs (Supplementary Fig. 3C, D, Supplementary Fig. 4C, D), demonstrating that crotonate exerted an impact on histone Kac process via ACSS2 effect. The SIRTs and HDACs were successfully overexpressed in TCMK-1 and HEK-293T cells after transfection with various plasmids (Supplementary Fig. 3E–P). Overexpression of SIRT1/2/3 and HDAC1/2/3 reduced the level of H3K9cr and H3K9ac in TECs (Supplementary Fig. 4E–H); however, overexpression of SIRT4/5/6 only affected H3K9ac modification without affecting H3K9cr (Supplementary Fig. 4I, J). Consequently, we proposed that ACSS2, notwithstanding H3K9ac interference, could be the ideal instrument for exploring the functions of H3K9cr in TECs and kidneys.

To understand the role of ACSS2-mediated H3K9cr in renal fibrosis, we generated mice with genetic deletion of ACSS2 using the CRISPR/Cas9 knock-out system (Supplementary Fig. 5A–C). The ACSS2$^{-/-}$ mice were born at the expected Mendelian ratio, with no birth or growth defects and no signs of kidney function defects. The results of western blotting confirmed that ACSS2 reduction in the kidneys of knockout mice (ACSS2$^{-/-}$) compared with littermate controls (Supplementary Fig. 5D, E). Interestingly, ACSS2 also decreased in fibrotic kidney of mice. The single-cell transcriptomics analysis from published studies suggested that ACSS2 was decreased in the proximal tubule (PT) S3 in the fibrotic kidney of mice[16,17], while increased in CKD patients[18] (Supplementary Fig. 6A, B). Mouse single-cell transcriptomics typically selects a piece of cortex for testing, while human kidney single-cell transcriptomics always uses intact cortical and medullary kidney penetrating tissue for subsequent experiments. The differences in single-cell transcriptome results indicate that different sampling methods have an impact on the expression pattern of ACSS2. On the other hand, the IHC staining revealed that ACSS2 was mainly expressed in the corticomedullary junction area of fibrotic kidneys, indicating a high correlation between ACSS2 expression and spatial characteristics (Supplementary Fig. 6 C). As the ability of single-cell

transcriptomics to distinguish tubular cells is far inferior to spatial transcriptomics (Supplementary Fig. 6D, E), we reanalyzed the spatial transcriptomics data from Dixon et al.[19]. It was revealed that PT S3 cells were those cells mainly expressed in corticomedullary junction with more expression of ACSS2 in fibrotic kidney compared to control (Supplementary Fig. 6F–H). Combined spatial transcriptomics data and our experiments, ACSS2 seemed to increase in special tubular epithelial cells of fibrotic kidneys while decreasing in bulk kidneys.

Next, two experimental kidney fibrotic mice (UUO and FAN) were created using ACSS2$^{-/-}$ mice and littermate controls. After evaluating the level of H3K9cr by western blotting, we confirmed that genetic knockout of ACSS2 could reduce the level of H3K9cr in fibrotic kidneys, which was consistent with the results of cell experiments. Similarly, the level of H3K9ac remained unchanged (Fig. 2B and Supplementary Fig. 9A, B). In addition, the knockout of ACSS2 also decreased the levels of H3K18cr, H3K14cr, and H3K23cr as well as H3K27ac, H3K4ac, and H3K36ac in UUO fibrotic kidneys (Supplementary Fig. 7), where similar results have been confirmed in TGFβ1-triggered epithelial tubular cells with transfected with ACSS2 plasmids (Supplementary Fig. 8). Furthermore, we analyzed the phenotype of these mice to explore the function of ACSS2 in kidney fibrosis (Fig. 2A). Histological changes, such as tubule atrophy and interstitial fibrosis, were alleviated in ACSS2$^{-/-}$ compared with wild-type (WT) UUO mice (Fig. 2C). The expression of protein markers of fibrosis including fibronectin (FN1), collagen type 6 (COL6), and smooth muscle actin (α-SMA) were higher in fibrotic kidneys, but lower in ACSS2$^{-/-}$ kidneys of UUO mice (Fig. 2D and Supplementary Fig. 9C–G). Transcript levels of fibrotic markers in kidneys including Fn1, collagen type 1a1 (Col1a1), and smooth muscle alpha (α)−2 actin (Acta2) were altered similarly to the levels of fibrotic markers at the protein level (Fig. 2E). Genetic deletion of ACSS2 also decreased H3K9cr level and alleviated kidney fibrosis in FAN mice (Supplementary Fig. 9H–K). All these data confirmed that global genetic deletion of ACSS2 could influence H3K9cr level and thus improve kidney fibrosis.

### Tubular-specific deletion of ACSS2 decreased H3K9 crotonylation to delay the progression of kidney fibrosis

TECs play a core role in kidney fibrosis[20]. The IHC staining results confirmed that ACSS2 was mainly expressed in TECs of fibrotic kidneys both in mice and human (Fig. 3A and Supplementary Fig. 10A). To investigate the contribution of ACSS2 in TECs to kidney fibrosis, we crossed ACSS2 flox mice with Ksp-Cre mice to selectively delete ACSS2 in TECs of the kidneys (Fig. 3B and Supplementary Fig. 10B–D). The Ksp-Cre ACSS2$^{fl/fl}$ (ACSS2$^{tecKO}$) mice and Cre-negative littermate controls (ACSS2$^{tecWT}$) were subjected to UUO or injected with folic acid (FA). The kidneys of ACSS2$^{tecKO}$ and ACSS2$^{tecWT}$ mice exhibited the same extent of H3K9ac when compared with sham-operated kidneys. The H3K9cr level was increased dramatically in ACSS2$^{tecWT}$ UUO kidneys, and was inhibited in kidneys of ACSS2$^{tecKO}$ UUO mice (Fig. 3C). The changes in H3K9cr and H3K9ac were also similar in the kidneys of FAN ACSS2$^{tecKO}$ mice (Supplementary Fig. 11A).

The results of H&E and Masson staining showed that tubule injury and collagen deposition were alleviated in the kidneys of ACSS2$^{tecKO}$ UUO mice compared with ACSS2$^{tecWT}$ UUO mice (Fig. 3D). The western blot and qPCR results also confirmed that kidney fibrosis of UUO and FAN mice were ameliorated in ACSS2$^{tecKO}$ mice compared with ACSS2$^{tecWT}$ mice, demonstrated by the decreased expression of fibrotic markers (Fig. 3E, F and Supplementary Fig. 11B–D). In summary, the data indicated that specific deletion of tubular ACSS2 decreased H3K9 crotonylation to delay the progression of kidney fibrosis.

### H3K9cr promoted cytokine production and regulated cytokine-cytokine receptor interaction in fibrotic kidneys

Since the genomic locations of H3K9cr have not been previously mapped in kidneys, it is difficult to explore the possible regulatory

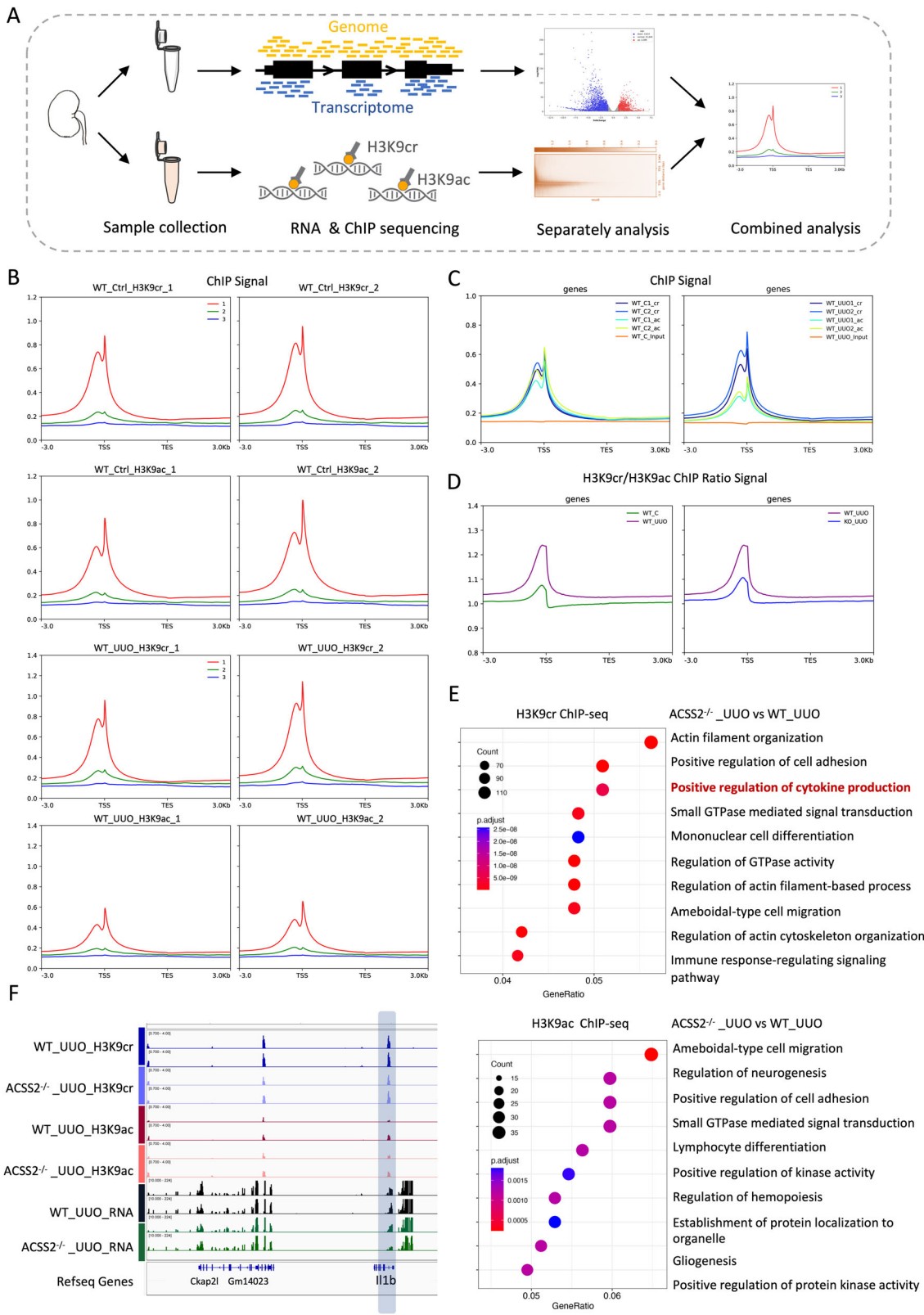

genes of H3K9cr through the public database. Therefore, we sought to determine the biological function of H3K9cr using chromatin immunoprecipitation sequencing (ChIP-seq). For comparison and as a control, we also performed H3K9ac ChIP-seq using the same samples (Fig. 4A). We found that both H3K9ac and H3K9cr were enriched at transcriptional start sites (Supplementary Fig. 12). The location of H3K9cr at transcriptional start sites is consistent with previous

findings[10,21]. Remarkably, the ChIP signal did not differ between H3K9cr and H3K9ac for control mice, while the ChIP signal of H3K9cr was obviously stronger than that of H3K9ac in the kidneys of UUO mice (Fig. 4C). The ChIP signal of H3K9cr and H3K9ac in control and UUO mice were shown in Fig. S13A.

We further performed RNA-seq on these samples, as the combination of ChIP-seq and RNA-seq data can be used to decipher the

**Fig. 4 | Integrated analysis of ChIP sequencing and RNA sequencing in kidneys of control and UUO from WT and ACSS2-/- mice. A** Outline of bioinformatic analysis strategy. **B** All genes were split into 3 equal groups based on their expression levels calculated from RNA-seq. Mean H3K9cr and H3K9ac ChIP signals are shown for each tertile of gene expression and shown as different colored lines (red line is the gene expressed highest, and blue line is the gene expressed lowest) in both control and UUO mice. **C** H3K9cr, H3K9ac, and input ChIP signals are shown in both control and UUO mice. TSS, transcriptional start site; TES, transcriptional termination site. **D** Mean H3K9cr to H3K9ac ChIP ratios in the control and UUO of WT and ACSS2$^{-/-}$ mice. **E** Comparable analysis from ChIP sequence between UUO of WT and ACSS2$^{-/-}$ mice using GO database, two-sided statistical tests. **F** Genome browser representation of RNA-seq reads and ChIP-seq reads for Il1b from UUO of control and ACSS2$^{-/-}$ mice. "Gene count" is the number of genes enriched in a GO (gene ontology) or KEGG (Kyoto Encyclopedia of Genes and Genomes) term. 'Gene ratio' is the percentage of total differential expression genes in the given GO/KEGG term. UUO unilateral ureteric obstruction, WT wild type, ACSS2$^{-/-}$: ACSS2 knockout.

transcriptional regulation network (Fig. 4A). Whether in control mice or UUO mice, the results of individual analyzes revealed a strong correlation between H3K9cr and H3K9ac regarding gene expression during the process of kidney fibrosis. The highest tertiles of gene expression displayed the highest occupancy of histone Kac and Kcr, suggesting that both H3K9cr and H3K9ac could activate gene transcription (Fig. 4B).

To further examine the relationship between H3K9cr and H3K9ac, we analyzed the relative ratio of H3K9cr to H3K9ac when both acylations can be detected (Fig. 4D). These results demonstrated that gene expression in UUO kidneys were among those with the highest H3K9 crotonylation/acetylation ratios. When deleting ACSS2 in UUO kidneys, the decrease in H3K9cr leads to lower H3K9cr/ac ratios. Nevertheless, deletion of ACSS2 exerted an insignificant effect on genes in control kidneys, as the H3K9cr level remained unchanged (Supplementary Fig. 13B). According to these findings, H3K9cr may activate gene expression similarly to H3K9ac, exerting a greater impact on renal fibrosis-related genes.

To understand the specific regulatory genes of H3K9cr in kidney fibrosis, we investigated the RNA-seq data between control and UUO kidneys. Gene Ontology (GO) and Kyoto Encyclopedia of Genes and Genomes (KEGG) pathway studies revealed the considerably enriched pathways of cytokine-cytokine receptor interaction and a cytokine-mediated signaling pathway in fibrotic kidneys (Supplementary Fig. 13C, D). After ACSS2 deletion in mice, the number of genes enriched in a pathway decreased (Supplementary Fig. 13E, F). Furthermore, to determine whether the reduction of the two signaling pathways produced by ACSS2 deletion is regulated by H3K9cr or H3K9ac, we investigated the results of GO pathway analysis from ChIP-seq data. It is noteworthy that the deletion of ACSS2 exhibited a greater impact on the genes controlling cytokine production by H3K9cr than H3K9ac in UUO kidneys (Fig. 4E).

Within these pathways, interleukin-1 (IL-1β) was found to be a core and altered proinflammatory cytokine, as mentioned in a previous study[16]. According to the enrichment plots of the GSEA results, the number of genes enriched in the pathways "cytokine-cytokine receptor interaction" and "response to IL-1" were increased in UUO kidneys while decreased after ACSS2 deletion (Supplementary Fig. 13G). It also raises the possibility that the deletion of ACSS2 offers renoprotection which mainly partially depends on H3K9cr-mediated IL-1β production. Excitingly, Il1b and Il1r1 displayed with increased H3K9cr level at their proximal promoter regions, and downregulated expression with ACSS2 deletion in UUO kidneys. However, these two genes displayed lower H3K9ac levels and fewer alterations after ACSS2 deletion (Fig. 4F and Supplementary Fig. 14A). The increase in Il1b and Il1r1 enrichment of H3K9cr following transfection of the ACSS2 plasmid in HEK-293T cells was further confirmed by ChIP-qPCR assay (Supplementary Fig. 14B). Interestingly, the enrichment of IL-1b also increased by H3K14cr, H3K27cr H3K36cr and H2BK34cr when we overexpress ACSS2 (Supplementary Fig. 14C). Overall, our findings indicated that H3K9cr, not H3K9ac, controlled the interaction between cytokines and cytokine receptors (particularly Il1b, which was crucial for fibrotic kidneys). However, whether crotonylation levels of other residues are also involved in regulating Il1b still needs further study. In addition, we analyzed the top five motifs from H3K9cr, which were HNF1b, COUP-

TFII, ERRg, HNF4a, HNF1 in control kidneys, and ERG, ETV2, ETS1, GABPA, Fli1 in UUO kidneys (Supplementary Fig. 14D).

## H3K9 crotonylation promoted IL-1β production in both kidney cells and fibrotic kidneys

As abovementioned, ChIP investigations indicated that IL-1β could be directly regulated by H3K9 crotonylation. We independently evaluated H3K9cr-mediated changes of IL-1β production in mice and cells (Fig. 5A). First, IL-1β expressed more in CKD patients and negatively correlated with eGFR (Supplementary Fig. 15A, B). In addition, the increase of IL-1β was accompanied by an increase level of H3K9cr in fibrotic kidneys of UUO and FAN mice (Fig. 5B and Supplementary Fig. 16A, B). From the spatial transcriptomics data, the increase of ACSS2 in PT S3 was also accompanied with the slight rise of IL-1β (Supplementary Fig. 15C). When global or tubular specific knockout of ACSS2 were applied to decrease H3K9cr modification in mice with kidney fibrosis, both the protein level and transcription of IL-1β were suppressed (Fig. 5D, E and Supplementary Fig. 16C, D). Remarkably, it was clear that the deletion of ACSS2 reduced the amount of IL-1β produced in UUO kidneys, without influencing serum concentration (Fig. 5C). Immunofluorescence co-staining of IL-1β and H3K9cr also revealed that those tubular epithelial cells which highly expressed H3K9cr (yellow) could be co-stained with IL-1β (red) (Fig. 5F and Supplementary Fig. 15D).

After demonstrating the alterations of H3K9cr-mediated IL-1β in vivo, we used two different kidney cell lines to confirm the link between H3K9cr and IL-1β in vitro. Both the protein and mRNA levels of IL-1β notably increased following crotonate stimulation (Supplementary Fig. 17A−D). The expression of IL-1β was further boosted by transfecting TCMK-1 cells with the ACSS2 plasmid, which raises H3K9cr level. In addition to intracellular IL-1β, the ELISA kit analysis showed that the concentration of IL-1β in the supernatant of ACSS2-overexpressed TCMK-1 cells was also elevated (Supplementary Fig. 17E−G) and the HEK-293T cells showed similar effects (Supplementary Fig. 17H−J). Furthermore, the primary tubular epithelial cells treated with TGFβ1 from WT and ACSS2$^{-/-}$ mice also linked the relationship between ACSS2 and H3K9cr-Il1b axis (Supplementary Fig. 17K−M). When SIRT1/2/3 and HDAC1/2/3 were overexpressed in TCMK-1 and HEK-293T cells to block H3K9cr modification, the levels of IL-1β decreased (Supplementary Fig. 18). Naturally, IL-1β expression was unaffected by SIRT4/5/6 plasmids, which had no impact on H3K9cr alteration (Supplementary Fig. 19). Collectively, our findings indicate that whether in vivo or in vitro, H3K9cr level is significantly associated with IL-1β production; importantly, ACSS2 may be a key factor to regulate H3K9cr-mediated IL-1β production.

## H3K9cr-derived IL-1β triggered macrophage activation in cells and fibrotic kidneys

As known, uncontrolled macrophages (particularly M1 macrophages) are considered deleterious in kidneys, as they sustain the proinflammatory environment, leading to the progression of renal injury and fibrosis[22]. IL-1β is an important cytokine that can induce proinflammatory or activated M1 macrophages. According to our spatial transcriptome analysis, TECs with the increase of ACSS2 were close to macrophage (Supplementary Fig. 20A-B); thus, it is possible that

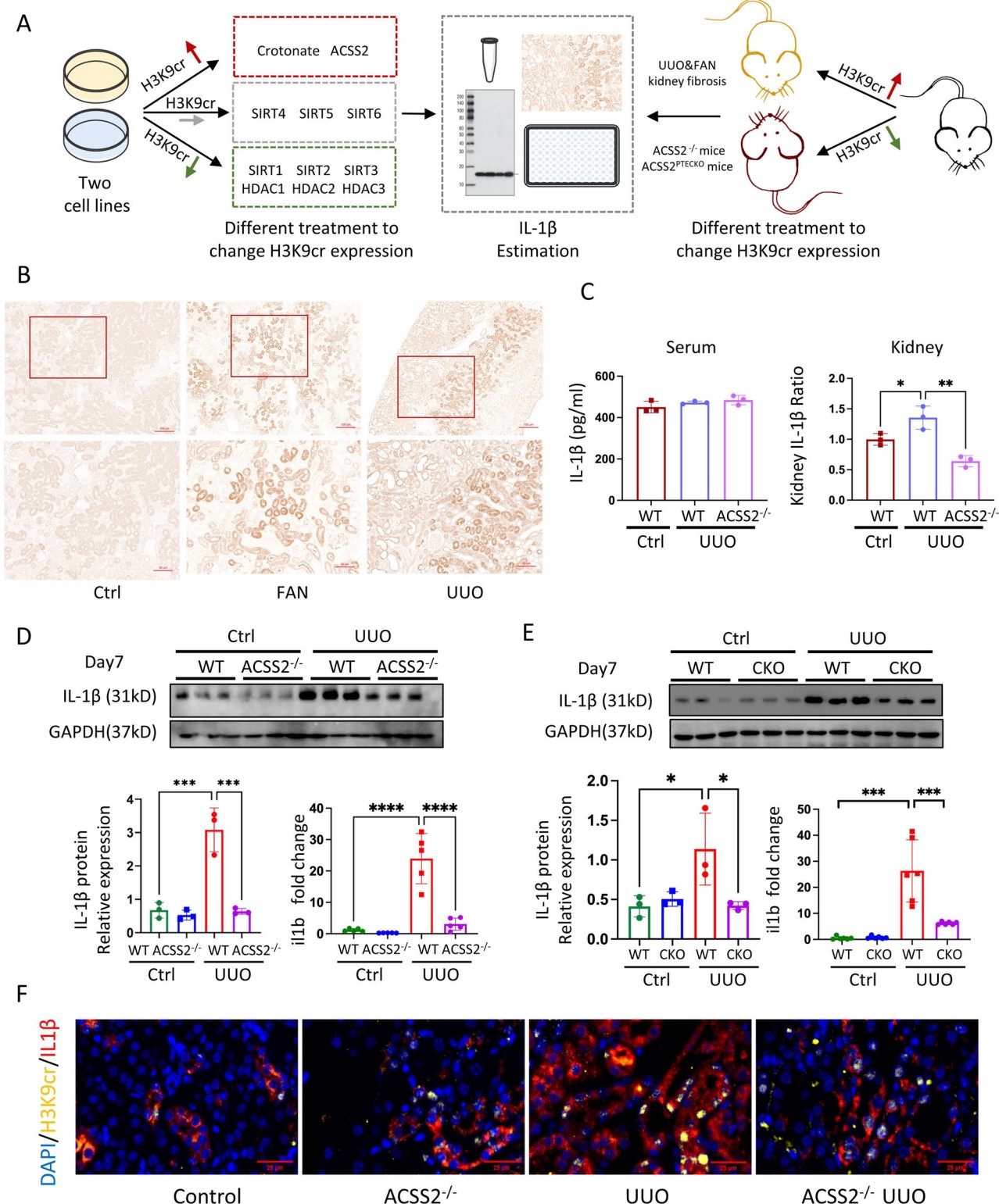

**Fig. 5 | The changes of IL-1β expression under different condition. A** Schematic overview of process for screening changes of IL-1β both in vivo and in vitro. **B** Representative IHC staining of IL-1β in kidneys of control and mice treated with UUO or injected with FA ($n = 3$ per group). Scale bar: upper panels: 100 μm; lower panels:50 μm. **C** IL-1β concentration of serum and kidney in control mice and UUO of WT and ACSS2$^{-/-}$ mice tested by ELISA kit ($n = 3$ per group, Control vs UUO: $p = 0.039$; UUO vs. ACSS2$^{-/-}$ + UUO: $p = 0.0042$). **D** IL-1β immunoblotting and quantification of IL-1β immunoblots normalizing to GAPDH ($n = 3$ per group, Control *vs.* UUO: $p < 0.001$; UUO vs. ACSS2$^{-/-}$ + UUO: $p < 0.001$), as well as IL-1β mRNA levels ($n = 5$ per group, Control vs. UUO: $p < 0.0001$;UUO vs ACSS2$^{-/-}$ + UUO: $p < 0.0001$). in kidneys of control and UUO from WT and ACSS2$^{-/-}$ mice. **E** IL-1β

immunoblotting and quantification of IL-1β immunoblots normalizing to GAPDH ($n = 3$ per group, Control vs. UUO: $p = 0.0342$; UUO vs. ACSS2CKO + UUO: $p = 0.0315$), as well as IL-1β mRNA levels ($n = 6$ per group, Control vs. UUO: $p < 0.001$; UUO vs ACSS2CKO + UUO: $p < 0.001$). in kidneys of control and UUO from WT and ACSS2$^{tecKO}$ mice. **F** IF colocalization staining of kidney H3k9cr (yellow), IL-1β (red), and DAPI (blue) in UUO of WT and ACSS2$^{-/-}$ mice ($n = 3$ per group). Scale bar: 25 μm. Ctrl control, UUO unilateral ureteric obstruction, WT wild type, ACSS2$^{-/-}$ ACSS2 knockout, ACSS2 CKO tubular epithelial cell-specific knock out of ACSS2, IHC immunohistochemical, IF immunofluorescence. Data shown are means ± SEM. Statistical analysis by one-way ANOVA with Tukey's post hoc test. *$p < 0.05$, **$p < 0.01$, ***$p < 0.001$ and ****$p < 0.0001$.

H3K9cr-mediated IL-1β triggers macrophage activation to promote kidney fibrosis progression. To test whether IL-1β could promote macrophage polarization, we first stimulated RAW264.7 macrophage with different doses of IL-1β. According to the Cell Counting Kit-8 assay, low dose of IL-1β can promote macrophage proliferation (Supplementary Fig. 20C). As expected, microscopic images depicted changes in cell morphology from a rounded M0 to flat M1 phenotype after IL-1β simulation (Supplementary Fig. 20D). Additionally, there were significantly higher levels of M1 macrophage markers (Tnf-α, iNOS, and Il1b), and lower levels of M2 markers (CD206) in IL-1β-simulated versus M0 cells, which this was consistent with the observed morphological changes (Supplementary Fig. 20E).

To test whether H3K9cr-derived IL-1β exerted similar effects, we collected the supernatant of HEK-293T cells transfected with ACSS2 plasmids ahead of time to increase H3K9cr level, and subsequently stimulated RAW264.7 macrophages (Fig. 6A). After supernatant stimulation, the microscopic images and mRNA levels of M1 macrophage markers including iNOS, Tnf-α, Mcp1, and Il6 changed, indicating that macrophage polarization occurred (Fig. 6B and Supplementary Fig. 21A). We repeated the experiments with TCMK-1 cells, and observed similar results (Supplementary Fig. 21B). In addition to supernatants from ACSS2-overexpressed cells, we collected supernatants from HEK-293T and TCMK-1 cells transfected with SIRT1/2/3 and HDAC1/2/3 plasmids, which inhibited H3K9cr-mediated IL-1β production partially. The microscopic images revealed that the flat M1 phenotype in the H3K9cr-downregulated group was less than those in the control group (Supplementary Fig. 22A). As the previous qPCR tests revealed, H3K9cr downregulation can reduce M1 macrophage markers (Supplementary Fig. 22B, C).

When observing the M1 macrophage marker in vivo, we found that regardless of whether global or tubular epithelial-specific knockout of ACSS2 was performed to decrease H3K9cr modification in mice with kidney fibrosis, the Cd68 and Tnf-α were suppressed (Supplementary Fig. 21C, D). Additionally, we also co-staining IL-1β with macrophage marker (F4/80) and a M1 macrophage marker (iNOS) (Supplementary Fig. 21E). Interestingly, IL-1β is mainly expressed in tubular epithelial cells instead of macrophage, which has been confirmed that just a few cells were co-stained with IL-1β and macrophage markers. Instead, it could be more colocalization of Il1r1 with macrophage markers (Supplementary Fig. 21F). Overall, these data suggest that regulating H3K9cr modification in TECs could influence the secretion of IL-1β, thereby influencing macrophage activation.

### H3K9cr-derived IL-1β accelerated tubular cell senescence in fibrotic kidneys

Other than mediating macrophage polarization, considerable studies report that IL-1β may be involved in the senescence of several types of cells, including vascular smooth muscle cells[23], astrocytes[24], bovine oviduct epithelial cells[25], mature chondrocytes[26], etc. Recently, several studies have shown positive correlations between senescent cell accumulation and fibrosis in kidneys during ageing[27–29] and disease[30–32]. Based on known results, we suspected that IL-1β—the key mediator identified by H3K9cr—could also regulate the senescence of TECs in kidney fibrosis. First, we administered IL-1β directly to TECs at different concentration to determine whether it affected tubular cell senescence. High doses of exogenous IL-1β did not cause any toxicity in TCMK-1 cells as shown in Fig. S23A; however, even low dose of IL-1β triggered senescence-associated β-galactosidase (SA-β-gal) positive cells (Supplementary Fig. 23B). IL-1β also increased the cellular senescence marker P53, and several senescence-associated secretory phenotype (SASP) markers (Il6, Mmp9, and Il1b) (Supplementary Fig. 23C).

To further confirm whether H3K9cr-derived IL-1β exert similar effects, we collected supernatants from TCMK-1 or HEK-293T cells transfected with ACSS2 plasmids, which are IL-1β enriched. Thereafter, we treated TCMK-1 cells to observe changes in senescent markers.

Similarly, the senescence phenotype increased in vitro, as evidenced by increased P53, Il6, and Mcp1 levels (Supplementary Fig. 24A, B). When we evaluated the data in vivo, the expression of P53, Il6, and Mcp1 increased in the kidneys of UUO and FAN mice (Supplementary Fig. 24C–F). Global knockout of ACSS2 in UUO and FAN mice to decrease H3K9cr partially mediated IL-1β production resulted in a decrease of P53, Il6, and Mcp1 (Supplementary Fig. 24G–J). Even when we specifically deleted ACSS2 in TECs, we discovered that the kidney cellular senescence marker decreased significantly (Supplementary Fig. 24K, L). The γH2AX staining to assess the DNA damage was also conducted, where the rise of DNA damage in UUO mice or primary tubular epithelial cells treated with Tgfβ will be suppressed when knockout or inhibition of ACSS2 (Supplementary Fig. 25A, B). These data therefore suggest that regulating H3K9cr level in TECs could influence the secretion of IL-1β and thus control the senescence of TECs.

### Anti-IL-1β IgG treatment alleviated macrophage activation and tubular cell senescence against fibrotic kidneys

We attempted to identify approaches to suppress IL-1β after confirming that H3K9cr-generated IL-1β might drive macrophage activation and tubular cell senescence. When anti-IL-1β IgG was added to RAW264.7 cells treated with IL-1β, the IL-1β-induced M1 macrophage polarization, cellular senescence marker P53, and SASP were also diminished (Fig. 6C, E). When treating cells with IL-1β antibodies to neutralize IL-1β-enriched supernatant stimulation, the changes of morphology and mRNA expression of M1 phenotypic markers and cellular senescence could be alleviated (Fig. 6D, F).

To examine the effects of IL-1β in vivo, we also administered anti-IL-1β IgG to UUO mice (Fig. 7A). Anti-IL-1β IgG dramatically improved renal fibrosis in UUO mice, as demonstrated by Masson's trichrome staining and the decreased mRNA and protein expression of kidney fibrotic markers (Fig. 7B–D and Supplementary Fig. 26A). Anti-IL-1β IgG treatment also suppressed the key markers of M1 macrophage and cellular senescence (Fig. 7E–G and Supplementary Fig. 26B, C). In addition, we treated UUO mice with ACSS2 inhibitor followed by IL-1β stimulation for rescue experiments in vivo. The injection of ACSS2 inhibitor can alleviate fibrosis, cellular senescence, and inflammation in UUO kidneys while IL-1β could eliminate these benefits (Supplementary Figs. 27, 28). To summarize the data presented above, anti-IL-1β IgG could alleviate M1 macrophage polarization and tubular cellular senescence caused by H3K9cr-mediated IL-1β both in vivo and in vitro, and thus improve kidney fibrosis.

### Pharmacological inhibition of ACSS2 partially repressed H3K9cr-mediated IL-1β production to protect against kidney fibrosis

Although anti-IL-1β IgG can help treat renal fibrosis, monoclonal antibodies are seemed to be prohibitively expensive for CKD patients who require long-term treatment. Another option is to use small-molecule chemical inhibitors, which could be less expensive and easier to use[33]. Unfortunately, no small-molecule IL-1β inhibitor is available on the market. Importantly, the newly developed ACSS2 inhibitor could serve as an alternative to inhibit IL-1β by reducing H3K9cr modification. In our study to discover a potential therapy, we identified an ACSS2 inhibitor (VY-3-249)[34] that could delay the progression of kidney fibrosis in mice (Fig. 8A). In addition, there are no differences in kidney and liver function between control and VY-3-249 administration group (Supplementary Fig. 29A). The tissue staining revealed that no obvious damage of heat, lung, liver, and spleen in VY-3-249 administration group with appropriate pharmacokinetic parameters (Supplementary Data 3 and Supplementary Fig. 29B).

ACSS2 expression was inhibited in kidneys of control mice after seven days of ACSS2 inhibitor administration (Supplementary

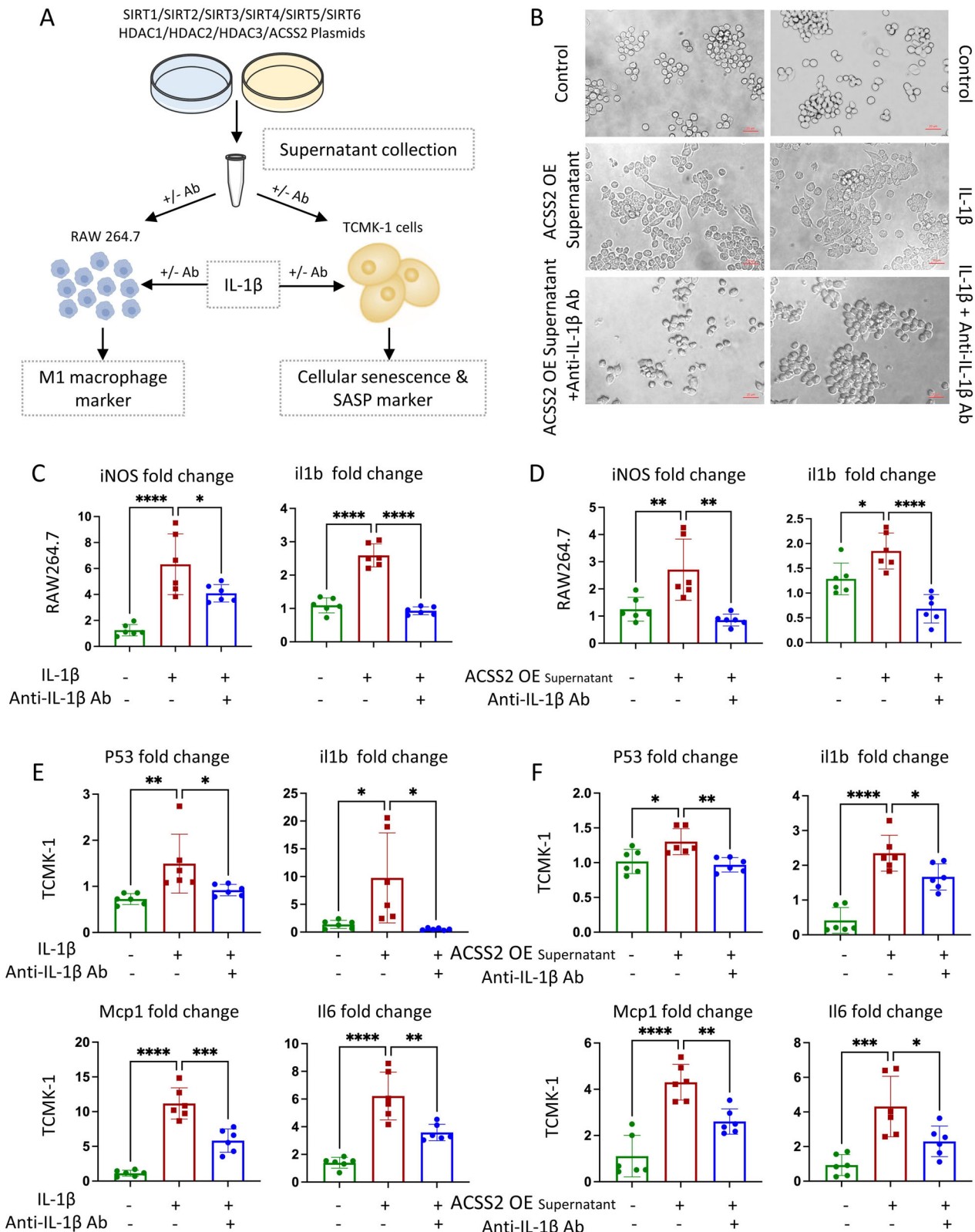

Fig. 30A). The ACSS2 inhibitor suppressed H3K9cr level, and had no effect on H3K9ac (Fig. 8B and Supplementary Fig. 30B, C). Remarkably, the ACSS2 inhibitor significantly improved kidney fibrosis in UUO and FAN mice as evidenced by the improvements in pathological staining, western blot, and qPCR for fibrotic markers (Fig. 8C–F and Supplementary Fig. 30D–G). As expected, IL-1β decreased after ACSS2 inhibitor treatment, accompanied by a decline in SA-β-gal-positive

senescent cells (Fig. 8F, G and Supplementary Fig. 31A–E). Additionally, the cellular senescence marker P53 and SASPs were suppressed (Fig. 8H and Supplementary Fig. 31F–J). Interestingly, these SASPs could also be identified by M1 macrophages, specifically Il1b, Il6, and Mcp1. The decrease in these proinflammatory markers was accompanied by a decrease of the M1 macrophage marker Cd86 (Fig. 8H).

**Fig. 6 | H3K9cr-derived IL-1β triggered M1 macrophage polarization and tubular senescence, which could be neutralized by anti-IL-1β IgG. A** Schematic overview of process for researching effects of H3K9cr-derived IL-1β on macrophage and tubular cells. **B** Microscopic images depicted the changes of RAW264.7 cells morphology after stimulation from IL-1β (5 ng/ml) or cellular supernatants collected from HEK-293T cells transfected with ACSS2 plasmids. IL-1β antibody (5 μg/ml) added to neutralize supernatants or IL-1β inverted the morphological changes. Scale bar: 20 μm. **C** iNOS and IL-1β mRNA levels of RAW264.7 which was stimulated with IL-1β with or without IL-1β antibody neutralization (number of cell wells = 6 per group, control vs IL-1β: $p < 0.0001$ for iNOS, $p < 0.0001$ for il1b; IL-1β vs IL-1β+Anti-IL-1β Ab: $p = 0.0406$ for iNOS, $p < 0.0001$ for il1b). **D** iNOS and IL-1β mRNA levels of RAW264.7 which was stimulated with cellular supernatants from HEK-293T cells transfected with ACSS2 plasmids with or without IL-1β antibody neutralization (number of cell wells = 6 per group, control vs supernatants: $p = 0.0076$ for iNOS, $p = 0.0228$ for il1b; supernatants vs supernatants+Anti-IL-1β Ab: $p = 0.0011$ for iNOS, $p < 0.0001$ for il1b). **E** P53, IL-1β, Il6, and Mcp1 mRNA levels of TCMK-1 cells which was stimulated with IL-1β (5 μg/ml) with or without IL-1β antibody (5 μg/ml) (number of cell wells = 6 per group, control vs IL-1β: $p = 0.0089$ for p53, $p = 0.0193$ for il1b, $p < 0.0001$ for Mcp1, $p < 0.0001$ for il6; IL-1β vs IL-1β+Anti-IL-1β Ab: $p = 0.0498$ for p53, $p = 0.0102$ for il1b, $p < 0.001$ for Mcp1, $p = 0.002$ for il6). **F** P53, IL-1β, Il6 and Mcp1 mRNA levels of TCMK-1 cells stimulated with cellular supernatants from 293 T cells transfected with ACSS2 plasmids with or without IL-1β antibody neutralization (number of cell wells = 6 per group, control vs supernatants: $p = 0.0191$ for p53, $p < 0.0001$ for il1b, $p < 0.0001$ for Mcp1, $p < 0.001$ for il6; supernatants vs. supernatants+Anti-IL-1β Ab: $p = 0.0068$ for p53, $p = 0.0367$ for il1b, $p = 0.0038$ for Mcp1, $p = 0.025$ for il6). ACSS2 OE ACSS2 overexpression. Data shown are means ± SEM. Statistical analysis by one-way ANOVA with Tukey's post hoc test. *$p < 0.05$, **$p < 0.01$, ***$p < 0.001$ and ****$p < 0.0001$.

## Discussion

Epigenetic regulation—especially histone modification—of gene expression plays key roles in cell fate transition in various diseases, including kidney fibrosis[35]. However, the newly reported histone Kcr changes and how they affect transcriptional responses in fibrotic kidneys remain poorly understood. In this study, the main findings were that overall, the increasing H3K9cr level was harmful to kidney fibrosis; the degree of H3K9cr could be modified by ACSS2 to further regulate IL-1β-mediated macrophage activation and tubular cell senescence. Our study demonstrates the potential of the therapeutic manipulation of histone Kcr by ACSS2 inhibition in a kidney fibrotic state.

Despite sharing the same DNA, the kidneys comprise complex tissues of multiple different cell types owing to differential epigenetic modulations that determine the characteristics of each cell type[36]. There is increasing interest in the epigenetic regulation of kidney injury and fibrosis, especially from the viewpoint of chronicity and aging[37]. Constitutive histone Kcr is present in healthy kidneys[8], while increased histone Kcr has been described in ADPKD[38] and AKI induced by FA and cisplatin[9]. As studies of Kcr in kidneys are rare and immature, the role of Kcr has not been unified. In kidneys with ADPKD, the increased Kcr could be reduced by genetic overexpression of CDYL and thereby slow cystic growth[38]. However, in FA-induced AKI, crotonate increased histone Kcr level and peroxisome proliferator-activated receptor coactivator alpha (PGC-1α) expression, thus providing protection against AKI[9]. One of the reasons for the difference in kidney protection or damage effects of histone Kcr may be that researchers use different ways to regulate histone Kcr levels. As we repeatedly emphasized, the same set of enzyme systems are shared by histone Kac and Kcr[8,12,13,15]. In our study, we identified and characterized several interventions that regulated H3K9cr levels in kidneys: genetically changing SIRT/HDAC/ACSS2 expression and increasing crotonate substrate availability. We found that several factors, including crotonate but excluding ACSS2, could both influence H3K9cr and regulate H3K9ac level. Data related to histone Kac has not yet been reported in an AKI study; thus, whether the renoprotective effects of crotonate are based on Kcr or Kac is difficult to differentiate. Moreover, finding a suitable tool to regulate Kcr separately—regardless of whether ACSS2 deletion or CDYL overexpression mice[38] are selected—was challenging. The similar results suggest that histone Kcr was involved in kidney injury and downregulation of histone Kcr modification may alleviate kidney damage. In another study, histone Kcr-regulated gene expression was increased to promote diseases progression[39]. Collectively, histone Kcr is involved in regulating renal disease, while besides H3K9, whether other residues of histone Kcr play significant roles still needs further experiments.

The mechanism by which ACSS2 regulated more H3K9cr than H3K9ac is possibly due to the concentration of crotonyl-CoA and acetyl-CoA[15,40]. Since the intracellular crotonyl-CoA concentration is about 600- to 1000-fold lower than that of acetyl-CoA, the crotonyl-CoA concentration is likely to be the limiting factor in the crotonyl transfer reaction[15]. Thus, Kcr is more sensitive to any changes that alter cellular crotonyl-CoA levels by the expression/activity/location/interacting partners of crotonyl-CoA-producing enzymes[41]. Identification of signals and transcription factors that induce expression of crotonyl-CoA-producing enzymes, as well as the nuclear interacting partners of these enzymes, will be important to further dissect the underlying molecular mechanisms that regulate Kcr level during kidney fibrosis. In addition, ACSS2 might regulate other residues behave like H3K9 site in term of an increase in Kcr but not Kac level. Even though we have tested the modification of several histone residues, the large number and complicated correlations of these residues required further liquid chromatography with tandem mass spectrometry (LC-MS/MS) technology with antibody-verified experiments to explore the comprehensive relationship between ACSS2 and modification of different residues in future.

Given that histone PTMs impact the expression of multiple genes, it remains to be explored whether changes in the expression of these specific genes or other genes are the key drivers of the observed detrimental effect of H3K9cr. Owing to our ChIP-seq data and studies reporting the association between kidney injury and inflammation, cytokine IL-1β has become a subject of interest[42]. Previous studies reported that the inhibition of IL-1β by human recombinant IL-1 receptor antagonist *Anakinra* or antimurine IL-1β IgG could protect kidney inflammation and fibrosis; however, the mechanism is mainly limited to one type of cell[43,44]. Recently, a single-cell analysis revealed that inflammatory signaling connections were active in maladaptive TECs and myeloid cells, as well as among epithelial cells in fibrotic kidneys; IL-1β was proved to be the important hub gene[16]. According to our data, tubular cell-derived IL-1β could stimulate M1 macrophage polarization to build a microinflammatory environment, which continues promoting persistent inflammation to damage tubular cells. Simultaneously, tubular cell-derived IL-1β could induce cellular senescence of TECs, as previously found in other cells[24,45]. Senescent TECs then secreted a variety of SASPs that further mediated chronic inflammation and exerted profound effects on neighboring cells[46–48]. It is possible that the SASPs from senescent TECs might activate macrophages and induce their polarization towards the secretory M1 type to exacerbate the inflammatory environment, as observed in the liver[49]. The relationships between TECs and macrophages to maintain this vicious cycle of maladaptive repair in kidney fibrosis are complicated; however, IL-1β may be a potential central regulator[16]. In fact, our report does not provide detailed information about how IL-1β could trigger cellular senescence directly. Previous studies mentioned that the oxidative stress-dependent mechanism is how microglia-derived IL-1β induces p53 activation[50], and miR-103/107 expression is part of the p53 response triggered by secreted IL-1β that renders macrophages refractory to HIV-1 entry[51]. In particular, recent results of *Canakinumab* anti-inflammatory thrombosis outcomes study trial underscore the clinical efficacy of selective IL-1β inhibition in the

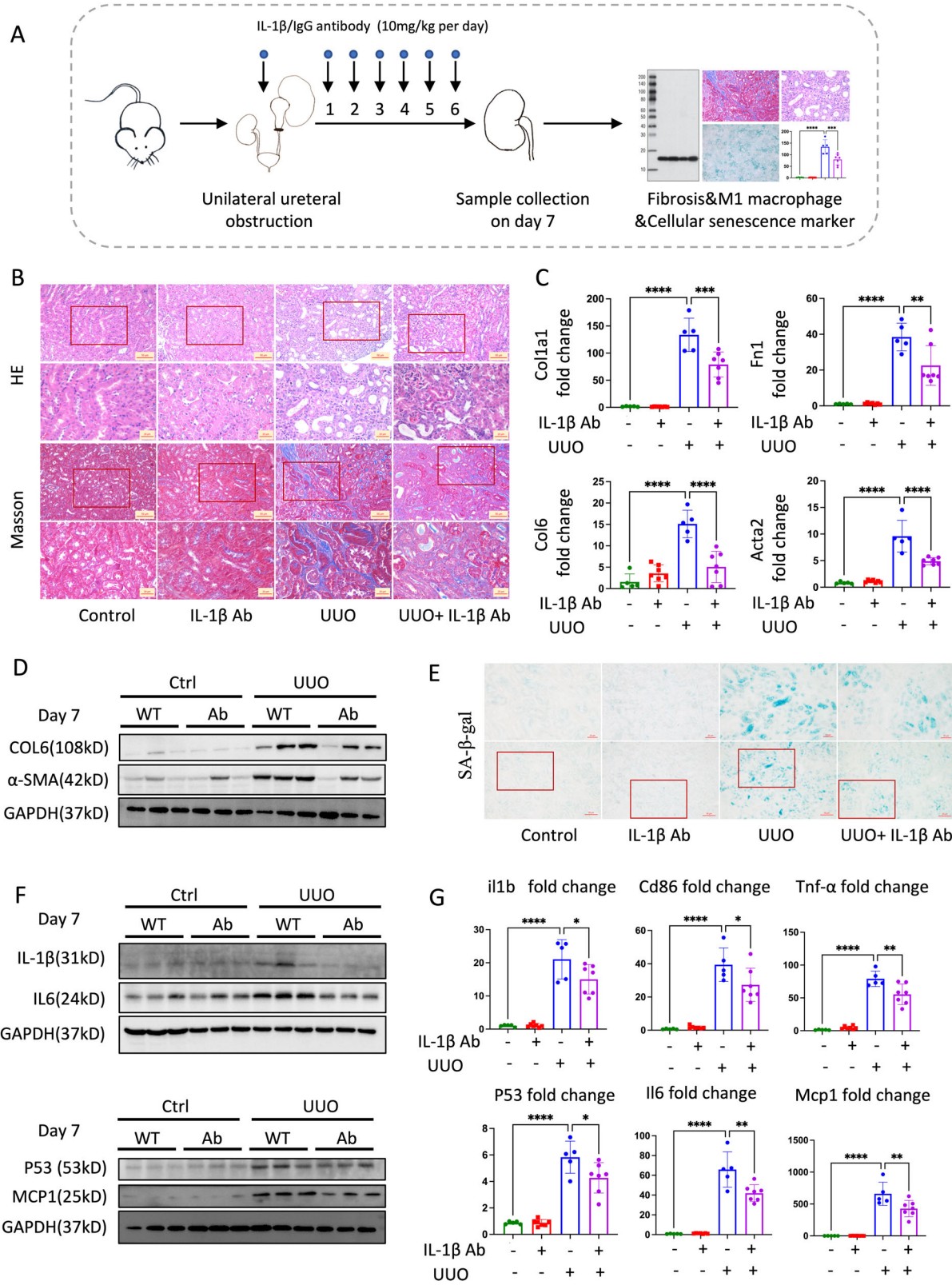

prevention of cardiovascular adverse events after acute coronary syndrome[24]. Perhaps the pharmacological inhibition of IL-1β will represent an effective therapeutic tool for both cardiovascular disease and kidney fibrosis.

In present, treatment of CKD is challenging, and the prognosis is difficult to determine. Our study identifies a potential clinical target to treat kidney fibrosis, besides the IL-1β mentioned above. The current

therapeutic drugs for PTMs have certain positive effects under various pathological conditions, including renal injury[52]. It is encouraging that some drugs targeting epigenetic modifier are already in clinical use or undergoing clinical trials, with some already reporting promising post hoc results (as is the case for apabetalone)[53,54]. However, most of our understanding of epigenetic modifiers and the kidneys is derived from the adverse effects of drugs already in clinical use for nonrenal

**Fig. 7 | Anti-IL-1β IgG inhibited cellular senescence, M1 macrophage markers and further alleviated kidney fibrosis in vivo. A** Treatment strategy of Anti-IL-1β IgG inhibitor for UUO mice. **B** Representative images of H&E and Masson staining in control and UUO mice treated with IL-1β antibody (10 mg/kg) (*n* = 3 per group).- Scale bar: upper panels: 50 μm; lower panels:20 μm. **C** Fn1, Col1a1, Col6 and Acta2 mRNA levels in the whole kidney lysates of control and UUO mice treated with IL-1β antibody (10 mg/kg) (number of control and UUO group is 5; number of IL-1β and UUO + IL-1β group is 7, control vs UUO: *p* < 0.0001 for Col1a, *p* < 0.0001 for Fn1, *p* < 0.0001 for Col6, *p* < 0.0001 for Acta2; UUO vs UUO+Anti-IL-1β Ab: *p* < 0.001 for Col1a, *p* = 0.0048 for Fn1, *p* < 0.0001 for Col6, *p* < 0.0001 for Acta2). **D** COL6, α-SMA, and GAPDH immunoblotting in the whole kidney lysates of control and UUO mice treated with IL-1β antibody (10 mg/kg) (*n* = 3 per group). **E** SA-β-gal staining in control and UUO mice treated with IL-1β antibody (10 mg/kg) (*n* = 3 per group).

Scale bar: upper panels: 20 μm; lower panels:50 μm. **F** IL-1β, IL6, P53, MCP1, and GAPDH immunoblotting in UUO mice neutralized with IL-1β antibody (*n* = 3 per group). **G** IL-1β, Cd86, Tnf-α, P53, Il6 and Mcp1 mRNA levels in the whole kidney lysates of control and UUO mice treated with IL-1β antibody (10 mg/kg) (number for control and UUO group is 5; number for IL-1β and UUO + IL-1β group is 7, control vs UUO: *p* < 0.0001 for il1b, *p* < 0.0001 for Cd86, *p* < 0.0001 for Tnf-α, *p* < 0.0001 for P53, *p* < 0.0001 for il6, *p* < 0.0001 for Mcp1; UUO vs UUO+Anti-IL-1β Ab: *p* = 0.0447 for il1b, *p* = 0.0404 for Cd86, *p* < 0.0001 for Tnf-α, *p* = 0.023 for P53, *p* = 0.0014 for il6, *p* = 0.007 for Mcp1,). UUO unilateral ureteric obstruction, WT wild type, Ctrl Control, Ab IL-1β neutralizing antibody. Data shown are means ± SEM. Statistical analysis by one-way ANOVA with Tukey's post hoc test. \**p* < 0.05, \*\**p* < 0.01, \*\*\**p* < 0.001 and \*\*\*\**p* < 0.0001.

indications[55,56]. The ACSS2 inhibitors may be another promising drug target to modify histone Kcr, which has been studied in several other diseases and has already demonstrated benefits[57-62]. Conversely, the success of clinical senotherapeutic trials in patients with kidney disease or HIV highlighted another way to treat CKD by alleviating cellular senescence[63,64]. The discoveries in our present study have unique clinical relevance and physiological significance. Nevertheless, this promising field merits further research that should focus on drugs already in clinical use or undergoing clinical trials, as they are likely to eventually translate these advances to clinical practice within a more reasonable time than completely new chemical entities.

Limitations of our work include uncertainty regarding the importance and transferability of the ACSS2-H3K9cr-IL-1β axis from animals to humans. Although we show the data of human kidney slides and cells, future studies need to confirm the importance of this axis using human kidney tissue. Another limitation is that the effect of pharmacological ACSS2 inhibition was examined using a single molecule. Testing multiple, these well-characterized ACSS2 inhibitors with distinct chemical scaffolds would be warranted to mitigate the risk of off-target effects. Although we have shown that alterations of H3K9cr are positively associated with changes in gene expression during kidney fibrosis, and further demonstrated that histone Kcr is functionally important for this process, we did not analyze in detail how this modification relates to known chromatin states (active versus poised versus heterochromatin), and do not explore the roles of other Kcrs in this paper. It is possible that crotonylation levels of other residue, not only H3K9, might be regulated by ACSS2 and could possibly play roles in regulating IL-1b-mediated renal fibrosis. Future LC-MS/MS analysis, ATAC sequencing and related experiments are needed to better understand the correlational and causal relationships between histone Kcr and overall chromatin remodeling, as well as the correlation between different Kcrs during kidney fibrosis. Last, the mechanism by which IL-1β is secreted by tubular cells and transferred to macrophages was not investigated in this paper. Based on prior studies, we suggest that exosomes—enriched with miRNAs and mRNAs[65]—might act as cargo for intercellular communication to provide the mechanism of cross-talk between TECs and macrophages[66]; still, further research is required.

In conclusion, we reveal that the pattern of histone crotonylation changes during kidney fibrosis, suggesting a role of H3K9cr in kidney injury and fibrosis. The degree of kidney cell histone Kcr can be therapeutically manipulated by ACSS2 inhibitors, and decreasing H3K9cr-mediated macrophage polarization and tubular senescence is nephroprotective. Collectively, our findings uncover that H3K9 crotonylation plays a critical, previously unrecognized role in kidney fibrosis, where ACSS2 represents an attractive target for strategies that aim to slow fibrotic kidney disease progression.

## Methods
### Animals
The protocol of the animal research was reviewed and authorized by the Experimental Animal Ethics Committee of West China Hospital, Sichuan University (2022527004). The C57BL/6 J mice (25–28 g) aged 6–8 weeks were purchased from GemPharmatech Co., Ltd. (Nanjing, Jiangsu, China). ACSS2 knockout mice and TEC-specific ACSS2 mice were acquired from Gempharmatech Co. Ltd. (Nanjing, Jiangsu, China). The mice were maintained in a temperature-controlled environment with a 12-hour light/dark cycle (lights-on at 7:00 a.m.) and had free access to food and water. All animals were randomly grouped (*n* = 3–7 mice per group). For unilateral ureteric obstruction (UUO) model, under anesthesia, the left ureter was isolated, and completely ligated with 3-0 silk suture at two points and cut between the ligatures to prevent retrograde urinary tract infection. The sham operation mice underwent an identical surgical intervention except for ureter ligation. Briefly, the abdominal cavity was exposed via a midline incision and the left ureter was isolated and ligated. The folic acid (FA) group was intraperitoneally injected with 250 mg/kg of folic acid which was diluted in 0.3 M sodium bicarbonate just once and the control mice received an injection of an equivalent volume of sodium bicarbonate alone.

### Human kidney biopsy slides
Human renal tissue, fixed in formaldehyde, and embedded in paraffin, was selected from the files of the Service of Pathology of West China Hospital, Sichuan University: control normal renal tissue was obtained from a patient with nephrectomy performed for neoplasia, involving the possibility of tumor-related immune exhaustion. Each patient gave informed consent before enrollment. The institutional ethical committee board approved the clinical protocol. Th research was performed according to the Helsinki's declaration principles.

### Drug studies for in vivo experiments
For the inhibitor studies, littermate mice were randomly injected with either the ACSS2 inhibitor or antimurine IL-1β antibodies. The ACSS2 inhibitor VY-3-249 (S8588, Selleck, Shanghai, China) was diluted in 0.9% saline and orally administered at a dose of 50 mg/kg/d for 7 consecutive days after UUO surgery or FA injection. Seven days after surgery or FA injection, the mice were sacrificed. Antimurine IL-1β antibody or control IgG was administered at 10 mg/kg body weight intraperitoneally once a day for 7 consecutive days after UUO surgery. Recombinant IL-1β was injected intraperitoneally at 1 μg/mouse, and ACSS2 inhibitor was administered 6 hours earlier, for a consecutive 7 days after UUO surgery. Mice were sacrificed on the 7th day after UUO surgery, and kidneys were harvested. All animal experiments were conducted in accordance with the Guidelines for the Care and Use of Laboratory Animals.

For the pharmacokinetic experiments, blood samples were collected from animals at 0, 0.083, 0.25, 0.5, 1, 2, 4, 6, 8, 10, and 24 h after ACSS2 inhibitor administration. Approximately 0.02 mL of blood samples were obtained from mice through the orbital venous plexus. Heparin sodium-containing sample tubes were used to immediately transfer the blood samples, which were then centrifuged at 3500 rpm for 10 min at 4 °C. The supernatant was collected as plasma. 100 μL of

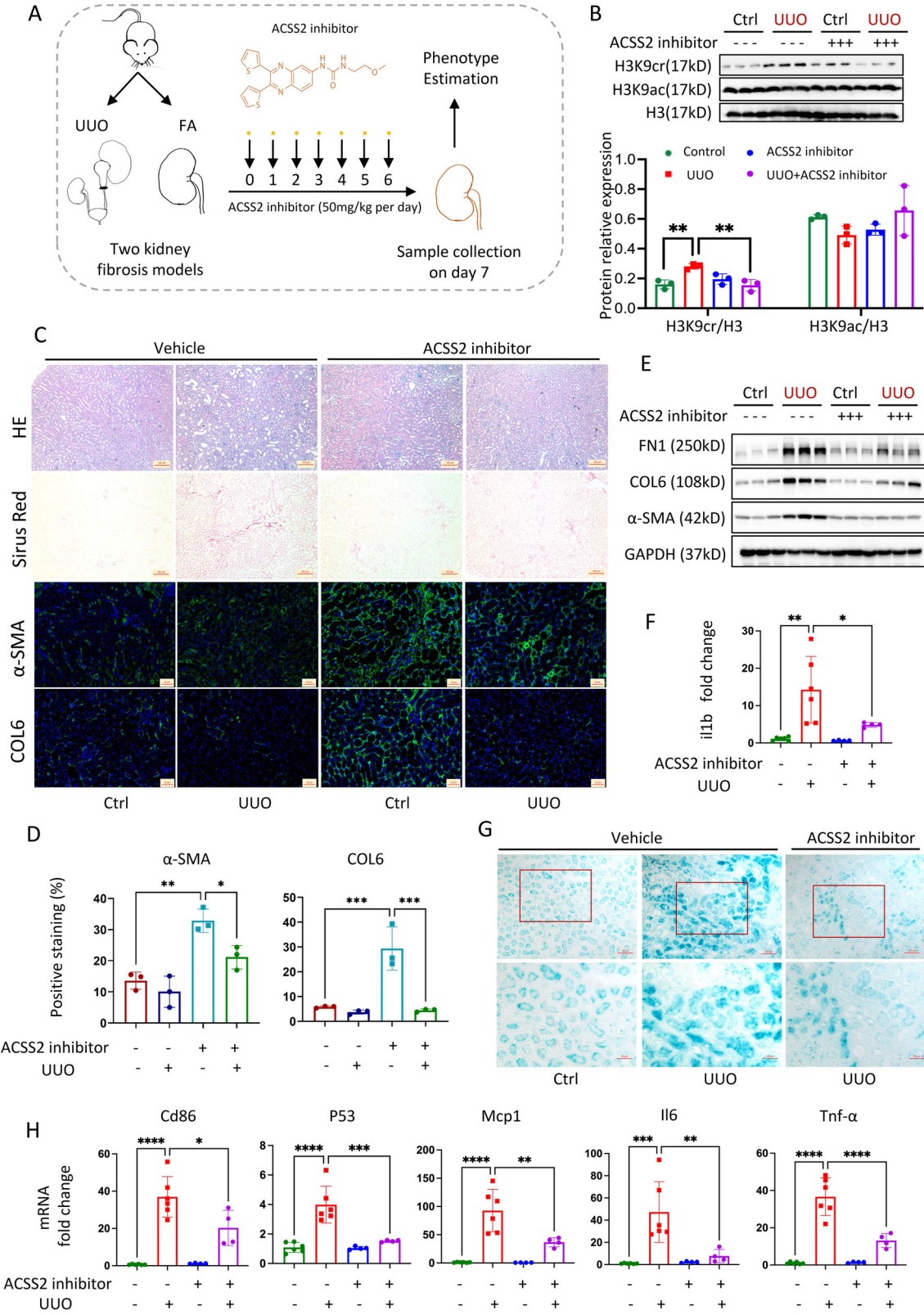

acetonitrile containing internal standard was added to 5 μL of plasma for protein precipitation, which were then centrifuged at 18900 g for 10 min at 4°C. The supernatant was transferred to sample tubes for UFLC-MS/MS analysis. The non-compartmental model of DAS2.0 software was applied to calculate the pharmacokinetic parameters after the plasma concentration was analyzed.

**Serum biochemistry assays**

Blood samples were taken and centrifuged at 1000 g for 20 min, and then serum was collected. The levels of serum creatinine (Scr), blood urea nitrogen (BUN), glutamic pyruvic transaminase (ALT) and glutamic oxaloacetic transaminase (AST) were measured by an automatic biochemical analyzer (Mindray BS-240, Shenzhen, China).

**Fig. 8 | ACSS2 inhibitor suppressed H3K9cr-mediated IL-1β expression, cellular senescence, and M1 macrophage markers as well as alleviated kidney fibrosis in vivo. A** Treatment strategy of ACSS2 inhibitor for mice. **B** Protein level of H3K9cr, H3K9ac, and H3 and quantification of these immunoblots in whole kidney lysates of UUO mice treated with or without ACSS2 inhibitor ($n = 3$ per group, control *vs.* UUO: $p = 0.0068$, UUO *vs.* UUO + ACSS2 inhibitor: $p = 0.0051$). **C** Representative images of H&E ($n = 6$ per group, Scale bar: 100 μm), Sirus red ($n = 3$ per group, Scale bar: 100 μm), and IF staining α-SMA (green), COL6 (green) and DAPI (blue) in UUO mice treated with or without ACSS2 inhibitor ($n = 3$ per group, Scale bar: 20 μm). **D** Quantitative analysis of α-SMA and COL6 IF staining in UUO mice treated with or without ACSS2 inhibitor ($n = 3$ per group, control vs. UUO: $p = 0.0014$ for α-SMA, $p = 0.0008$ for COL6; UUO *vs.* UUO + ACSS2 inhibitor: $p = 0.0254$ for α-SMA, $p = 0.0005$ for COL6). **E** FN1, COL6, α-SMA and GAPDH immunoblotting in the whole kidney lysates of UUO mice treated with or without ACSS2 inhibitor ($n = 3$ per group). **F** mRNA levels of IL-1β in whole kidney lysates of UUO mice treated with or without ACSS2 inhibitor (the number for control and UUO group is 6, the number for ACSS2 inhibitor and UUO + ACSS2 inhibitor group is 4; control vs. UUO: $p = 0.0016$, UUO vs UUO + ACSS2 inhibitor: $p = 0.04494$). **G** SA-β-gal staining of control, UUO fibrotic mice, and UUO mice treated with ACSS2 inhibitor ($n = 3$ per group). Scale bar: upper panels: 100 μm; lower panels:50 μm. **H** mRNA levels of Cd86, P53, Mcp1, Il6 and Tnf-α in whole kidney lysates of control, UUO fibrotic mice and UUO mice treated with ACSS2 inhibitor (number for control and UUO group is 6, number for ACSS2 inhibitor and UUO + ACSS2 inhibitor group is 4; control *vs.* UUO: $p < 0.0001$ for Cd86, $p < 0.0001$ for P53, $p < 0.0001$ for Mcp-1, $p = 0.0005$ for il6, $p < 0.0001$ for Tnf-α; UUO *vs.* UUO + ACSS2 inhibitor: $p = 0.0136$ for Cd86, $p = 0.0004$ for P53, $p = 0.0043$ for Mcp-1, $p = 0.0056$ for il6, $p < 0.0001$ for Tnf-α). Ctrl Control, UUO unilateral ureteric obstruction. Data shown are means ± SEM. Statistical analysis by one-way ANOVA with Tukey's post hoc test. *$p < 0.05$, **$p < 0.01$, ***$p < 0.001$ and ****$p < 0.0001$.

## Histologic examination

Remove half of the kidney, freeze in a liquid nitrogen and keep under a temperature of −80 °C. One-quarter of the kidney was excised and then fixed in 10% formaldehyde (50-00-0, Chron Chemicals, Chengdu, China), dehydrated, embedded into the paraffin and sectioned at a thickness of 4 μm for H&E, Masson and Sirius Red staining. An additional quarter of the kidney were embedded into OCT compound, which was frozen under a temperature of −80 °C. With Vectra® Polaris™ Automated Quantitative Pathology Imaging System and AxioCamHRc digital camera (Carl Zeiss, Jena, Germany), kidney sections were scanned and pictured at 100×, 200× and 400× magnification.

## Immunohistochemistry (IHC) staining analysis

The paraffin-embedded kidneys were sectioned to a thickness of 4 μm, de-paraffinized, rehydrated, and next antigen-retrieved. Subsequently, these sections were blocked with 2.5% normal goat serum, and inoculated by the primary antibodies Pan-Kcr, H3K9cr, ACSS2 or IL1b diluted 200:1 in PBS under a temperature of 4 °C overnight, separately. Slides were cleaned in PBS 3 times and stained with the VECTASTAIN ABC kit (Vector, Burlingame, CA, USA), which were then visualized through utilizing AxioCamHRc digital camera (Carl Zeiss, Jena, Germany) with ZEN 2012 microscopy software (blue version).

## Immunofluorescence (IF) staining analysis

Sections of OCT-embedded kidneys were 4 μm in thickness and subsequently inoculated at room temperature utilizing 5% horse serum for 60 min for blocking the non-specific binding sites. The slides were next inoculated overnight in a humid chamber under a temperature of 4 °C after dilution 1:200 in PBS with primary antibody Pan-Kcr, H3K9cr, α-SMA, COL6, γH2AX. The equivalent secondary antibody (1:500 dilution, 111-025-003, Jackson ImmunoResearch, West Grove, PA, USA) was utilized for 60 min. Slides were re-cleaned, stained with DAPI (dilution 1:500, D8200, Solarbio, Beijing, China), and sealed through coverslips. For cells, formaldehyde was used for fixation for 15 min, and the primary antibody γH2AX was added to the slides at 4 °C overnight. Secondary antibodies (1:500 dilution, 111-545-144, Jackson ImmunoResearch, West Grove, PA, USA) was applied for 60 min at room temperature. After washing, the cells were stained with DAPI (dilution 1:500, D8200, Solarbio, Beijing, China). The co-staining of IL-1β with H3K9cr, IL-1β with iNOS and F4/80 were performed by Opal reagent (Y6084S, Y6088S, Y6094S, Uelandy, Suzhou, China) according to the guidelines of the manufacturer. Images were gathered under an AxioCamHRc digital camera (Carl Zeiss, Jena, Germany) via utilizing the ZEN 2012 microscopy software (blue version).

## SA-β-gal staining

The activity of SA-β-gal was investigated in accordance with the guidelines of the manufacturer utilizing a kit (C0602, Beyotime Biotechnology, Shanghai, China). Images were gathered randomly with an AxioCamHRc digital camera (Carl Zeiss, Jena, Germany).

## Measurement of IL-1β by ELISA

Blood samples were centrifuged at 1000 g for 15 min and stored at −80°C until use. Frozen kidney tissues (10 mg) were added 100 μL PBS, homogenized with an electric homogenizer, centrifuged for 20 min at 13,000 rpm at 4 °C and stored at −80 °C until use. Pipette cell culture media into a centrifuge tube, centrifuge at 1,500 rpm for 10 min at 4 °C, and stored at −80 °C until use. Levels of IL-1β of mice serum, mice kidney tissue, and cell culture supernatant were detected by solid-phase sandwich enzyme-linked immunosorbent assay (ELISA) kits (Ruixin Biotechnology, Quanzhou, China) specific for these factors, and absorbance was measured at 450 nm using a plate reader (BioTek ELx800, USA).

## Western blotting

Nearly, 20-30 mg kidney tissue was homogenized in SDS-lysis buffer (#7722, CST) containing 42 mM DTT following the user manual. The same samples were loaded to the SDS-PAGE gels and run at 100 v for 1 h 40 min at RT in Tris-Glycin-SDS containing buffer. Blots were blocked with 5% non-fat dry milk powder in tris-buffer saline containing Tween-20 (TBST) for 30 min at RT. The blots were incubated in primary antibody overnight at 4 °C. After primary antibody (all the antibodies were shown in supplementary Data 4) incubation, blots were washed three times with TBST. Horseradish peroxidase-labeled goat anti-rabbit IgG (HA1001, 1:2000 dilution; HuaBio, Hangzhou, China) or goat anti-mouse IgG (HA1006, 1:2000 dilution; HuaBio, Hangzhou, China) were probed for 1 h at RT prepared in non-fat dry milk containing Tween-20. Finally, blots were washed with TBST for three minutes each 10 min at RT, and observed through Odyssey infrared imaging system. (Fluorescence Chemiluminescence Imaging System, Clinx Science, Shanghai, China) and quantified through utilizing ImageJ software (version 6.0; Wayne Rasband, National Institutes of Health, USA).

## Histone extraction, western blotting, and LC-MS/MS analysis

To extract histones from kidneys, we first isolated the nuclei and proceeded with the acid-histone extraction method. Two micrograms of histone lysates were loaded onto 15% SDS-PAGE gels; Coomassie brilliant blue was used for gel staining to demonstrate histone purification. Rest histone lysates were used for LC-MS/MS to double-check histone purification.

Briefly, nearly 40 mg of kidney tissue was washed with ice-cold PBS and minced in the nuclei isolation buffer (NIB-250 is 15 mM Tris-HCl at pH 7.5, 60 mM KCl, 15 mM NaCl, 5 mM MgCl$_2$, 1 mM CaCl$_2$, and 250 mM sucrose, to which 0.1% Nonidet P-40, 1x protease inhibitor cocktail, 1 mM DTT, and 10 mM sodium butyrate). Kidney pieces were collected into glass Dounce homogenizer on ice. After 5 min of

incubation on ice, the homogenates were spin down, nuclei pellet was washed twice with NIB-250 without NP-40 and proceeded with histone extraction.

Histones were extracted with 0.4 N $H_2SO_4$ at 5:1 ratio for 2 h at 4 °C. Acidified nuclei were spin down at 11000rcf for 10 min at 4 °C, and the soluble fraction containing histones was collected into new tube and precipitated with 20% trichloro acetic acid at final concentration overnight at 4 °C overnight. Samples were spin down at 11000rcf for 10 min at 4 °C to sediment the histone pellet at the bottom. Histone pellets were washed with ice-cold 1 mL acetone containing 0.1% 12 N HCl then the pellet was washed twice with ice-cold 100% acetone, air-dried, and dissolved in RIPA buffer.

### Histone liquid chromatography-mass spectrometry

Trypsin Digestion The sample was slowly added to the final concentration of 20% v/v TCA to precipitate protein, then vortexed to mix and incubated for 2 h at 4 °C. The precipitate was collected by centrifugation at $4500\,g$ for 5 min at 4 °C. The precipitated protein was washed with pre-cooled acetone for 3 times and dried for 2 h. The protein sample was then redissolved in 200 mM TEAB and ultrasonically dispersed. Trypsin was added at 1:50 trypsin-to-protein mass ratio for the first digestion overnight. The sample was reduced with 5 mM dithiothreitol for 30 min at 56 °C and alkylated with 11 mM iodoacetamide for 15 min at room temperature in darkness. Finally, the peptides were desalted by $C_{18}$ SPE column. LC-MS/MS Analysis: The tryptic peptides were dissolved in solvent A (0.1% formic acid, 2% acetonitrile/in water), directly loaded onto a home-made reversed-phase analytical column (25-cm length, 75/100 µm i.d.). Peptides were separated with a gradient from 6% to 22% solvent B (0.1% formic acid in acetonitrile) over 40 min, 22% to 30% in 12 min and climbing to 80% in 4 min then holding at 80% for the last 4 min, all at a constant flow rate of 450 nL/min on a nanoElute UHPLC system (Bruker Daltonics). The peptides were subjected to a capillary source followed by the timsTOF Pro (Bruker Daltonics) mass spectrometry. The electrospray voltage applied was 1.60 kV. Precursors and fragments were analyzed at the TOF detector, with an MS/MS scan range from 100 to 1700 m/z. The timsTOF Pro was operated in parallel accumulation serial fragmentation (PASEF) mode. Precursors with charge states 0 to 5 were selected for fragmentation, and 10 PASEF-MS/MS scans were acquired per cycle. The dynamic exclusion was set to 24 s. The peptide segment was dissolved in the mobile phase A of liquid chromatography and separated by the NanoElute ultra-high performance liquid phase system. Mobile phase A is an aqueous solution containing 0.1% formic acid and 2% acetonitrile; Mobile phase B is a solution containing 0.1% formic acid and 100% acetonitrile. Liquid phase gradient setting: 0–40 min, 6–22% B; 40–52 min, 22% ~ 30% B; 52-56 min, 30% ~ 80% B; 56–60 min, 80% B, flow rate maintained at 450nL/min. The peptide segment is separated by the ultra-high performance liquid phase system and injected into the Capillary ion source for ionization and then analyzed by the timsTOF Pro mass spectrometry. The ion source voltage is set at 1.6 kV, and the peptide parent ion and its secondary fragments are detected and analyzed using high-resolution TOF. The scanning range of secondary mass spectrometry is set to 100–1700. The data acquisition mode uses parallel cumulative serial fragmentation (PASEF) mode. After a primary mass spectrum is collected, the secondary spectrum with the charge number of the parent ion in the range of 0-5 is collected by 10 times in PASEF mode. The dynamic elimination time of the tandem mass spectrum scanning is set to 24 s to avoid repeated scanning of the parent ion. Database Search The resulting MS/MS data were processed using MaxQuant search engine (v.1.6.15.0). Tandem mass spectra were searched against the Mus_musculus_10090_-SP_20220119_Histone_H. fasta database (50 entries) concatenated with reverse decoy database. Trypsin/P was specified as cleavage enzyme allowing up to 4 missing cleavages. The mass tolerance for precursor ions was set as 20 ppm in first search and 20 ppm in main search, and

the mass tolerance for fragment ions was set as 0.02 Da. FDR was adjusted to <5%.

### RNA isolation and quantitative real-time PCR

Total RNA was extracted from kidney and spleen tissues using a total RNA extraction kit (Foregene, Chengdu, China) according to the protocols and was reverse transcribed into cDNA using a PrimeScript™ RT reagent kit (Takara, Kusatsu, Japan). The mRNA concentration was measured using a Scan Drop 100 (Analytik Jena, Thuringia, Germany) instrument. Quantitative real-time PCR was performed by using iQ SYBR Green Supermix (Bio-Rad, Hercules, CA, USA) in a PCR system (CFX Connect; Bio-Rad, Hercules, CA, USA). The primers used for target mRNA detection are listed in Supplementary Data 5. Relative gene expression was normalized to that of GAPDH by comparison with the control groups using CFX Manager™ Software (Bio-Rad, Hercules, CA, USA).

### RNA sequencing

Frozen kidney samples from groups ($n = 3$ per group) were chosen randomly for sequencing. With TRIzol reagent (Invitrogen, Carlsbad, CA, USA), the total RNA of the samples could be extracted, and then the samples were examined for purity, quality, and integrity. Through LC-BIO Bio-Tech Ltd (Hangzhou, China), the construction and sequencing of libraries were conducted. With Illumina NovaSeq 6000 platform, such libraries were subsequently sequenced and paired-end reads with a 2×150 bp read length were produced.

### Sample collection and preparation of ChIP sequencing

ChIP assays were performed by Shandong Xiuyue Biotechnology Co., Ltd, according to the standard crosslinking ChIP protocol with modifications. Briefly, cells were harvested and crosslinker with 1% formaldehyde for 10 min at room temperature. After sonication, immunoprecipitation was performed with anti-H3K9cr (PTM) and anti-H3K9ac (Active Motif). The immunoprecipitated complex was washed, and DNA was extracted and purified by Universal DNA Purification Kit (TIANGEN, #DP214). The ChIP-Seq library was prepared using original Ultra II DNA Library Kits (NEB, #E7645) according to the manufacturer's instructions. For ChIP-seq, extracted DNA was ligated to specific adapters followed by deep sequencing in the Illumina Novaseq 6000 using 150 bp paired-end.

### Data analysis of ChIP sequencing

Raw data (raw reads) of fastq format were firstly processed through in-house perl scripts. In this step, clean data (clean reads) were obtained by removing reads containing adapter, reads containing ploy-N and low-quality reads from raw data. At the same time, Q20, Q30 and GC content the clean data were calculated. All the downstream analyzes were based on the clean data with high quality. Clean reads were mapped to the reference genome using Bowtie2 software. And reads from organelle, mapping quality smaller than 30, PCR duplicated reads all removed. Those high-quality mapping reads were subjected to further peak calling. Macs2 was used to call peaks with q value < 0.05. Peaks were annotated by ChIP seeker package.

We analysis differential accessible peak through 3 steps. First, merge the peak files of each sample using the bed tools software. Second, the counts of the reads over the bed were determined for each sample using bed tools multicov. Finally, differential accessible peak was assessed using DESeq2. The region were called differentially accessible if the absolute value of the log2 fold change was 1 at an $p$ value < 0.05. Gene ontology (GO) analysis was performed to facilitate elucidating the biological implications of unique genes in the significant or representative profiles of the gene in the experiment [Ashburner M, et al. Gene ontology: a tool for the unification of biology. The Gene Ontology Consortium. Nat Genet. 2000 May;25(1):25-9.]. We downloaded the GO annotations from NCBI, UniProt (http://

www.uniprot.org/) and the Gene Ontology (http://www.geneontology.org/). Fisher's exact test was applied to identify the significant GO categories and FDR was used to correct the *p*-values. Pathway analysis was used to find out the significant pathway of the genes according to KEGG database. We turn to Fisher's exact test to select the significant pathway, and the threshold of significance was defined by *P*-value and FDR.

## ChIP-qPCR assay
Proteins and DNA interaction was evaluated by ChIP-qPCR using the ChIP assay kit (Millipore, MA, USA). The experiment protocols were according to the manufacturer's instructions. The antibodies used for the ChIP assay were as follows: anti-H3k9cr (PTM), anti-H3k9ac (Active Motif), anti-H3k4cr (PTM), anti-H3k14cr (PTM), anti-H3k18cr (PTM), anti-H3k23cr (PTM), anti-H3k27cr (PTM), anti-H3k36cr (PTM), anti-H2Bk34cr (PTM), and control IgG (Millipore). The primers used for ChIP were as follows: Il1b-F 5′- ACCCAGACAGGGCTTTTAGC-3′, Il1b-R 5′- TCCATTCCTAACACTGAGCCC-3′; Il1r1-F 5′- TCACTCAGGTCCTCT-CAGTCC-3′, Il1r1 -R 5′- TCCAATTGTGGGCAGCAATGA-3′. The calculation formula for enrichment efficiency was elaborated on previously.

## Isolation and culture of primary tubular epithelia cells
Renal cortex was isolated from mouse kidney and divided into tiny pieces, then added with 1 mg/ml type I collagenase (17100-017, Gibco, MA, USA), and incubated at 80 rpm in a 37 °C shaker for 30 min. Subsequently, PTCs were separated by strainer (100, 70, and 40 μm), and erythrocytes were removed with red blood cell lysis buffer (R1010, Solarbio, Beijing, China). Finally, the PTCs were cultured in RPMI1640 (HyClone, Logan, UT, USA) containing 10% fetal bovine serum, 1×insulin transferrin selenium additive, 1% penicillin/streptomycin and 20 ng/mL epidermal growth factor, and incubated at 37 °C in a humidified atmosphere of air/$CO_2$ (95:5).

## Cell viability assay
Cell viability was determined by the Cell Counting Kit-8 assay (CCK-8, APExBIO, Houston, TX, USA) according to the manufacturer's instructions. Briefly, TCMK-1 cells in the logarithmic growth phase were seeded in 96-well culture plates at a density of 5000 cells/well. After treating with Il1b, a 10 μl CCK-8 solution was added to each well and incubated in the dark for 1 h at 37 °C. In the end, the absorbance at 450 nm was detected using a microplate reader (Synergy Mx, Biotek, Vermont, USA).

## Cell culture and treatments
TCMK-1, HEK-293T, and RAW264.7 macrophage were acquired from the American Type Culture Collection (Manassas, VA). TCMK-1, HEK-293T and RAW264.7 cells were cultured in DMEM (Sigma-Aldrich) containing 10% fetal bovine serum (FBS) and incubated at 37 °C in a humidified atmosphere of air/$CO_2$ (95:5). For IL-1β-induced aging research, TCMK-1 and HEK-293T were incubated in MEM medium supplemented with different doses of IL-1β for 24 h. For assessing the treatment effect of IL-1β antibody, cells were treated with IL-1β (5 ng/ml) or cell culture supernatant with IL-1β antibody (5 μg/ml) at the same time for 24 h. For TGFβ-induced fibrosis research, HEK-293T and primary tubular epithelia cells were exposed to TGFβ (20 ng/ml) for 48 h. For plasmid transfection research, gene-expressing plasmid and blank plasmid (pVector) were obtained from MiaoLing Plasmid (Wuhan, China). TCMK-1 and HEK-293T cells transfection with plasmids was conducted using Lipofectamine 2000 (12566014, Invitrogen, CA, USA) for 24 h, according to the manufacturer's instructions.

## Primary culture of mouse TECs
Kidneys were collected from 3- to 6-week-old mice, minced, and digested by collagenase I (2 mg/ml) for 30 min at 37 °C and then filtered successively through 100-, 70-, and 40-μm mesh to collect single TECs. Cells were cultured in RPMI 1640 supplement with 10% fetal bovine serum (FBS), epidermal growth factor (20 ng/ml), and 100× insulin-transferrin-selenium at 5% $CO_2$, 37 °C. At 80% confluence, cells were treated in fed condition (RPMI 1640 + 10% FBS).

## Statistical analysis
Statistical analysis was performed using the GraphPad Prism 9 software. One-way ANOVA and the unpaired t-test were performed on variables. Sample size estimation was not performed, and sample size was determined by the number of animals in the colony of a determined age and gender. The number of replicates (including the number of animals used in each experiment) are indicated in the figures and/or figure legends. All data are expressed as mean ± SEM. The statistical parameters can be found in the figures and the figure legends. $p < 0.05$ was considered significant. $p < 0.05$ (*), $p < 0.01$ (**), $p < 0.001$ (***) and $p < 0.0001$ (****).

## Reporting summary
Further information on research design is available in the Nature Portfolio Reporting Summary linked to this article.

## Data availability
The sequencing data generated in this study have been deposited in the GEO data repository database under accession code GSE245390 and GSE253032. The remaining data are available within the article, Supplementary Information or Source Data file. Source data are provided with this paper, and have been deposited in Figshare database(https://doi.org/10.6084/m9.figshare.25202678).

## Code availability
The code of analyzes is available and provided in Source Data file, and have been deposited in Figshare database(https://doi.org/10.6084/m9.figshare.25202678).

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

## Acknowledgements

This work was supported by National Natural Science Foundation of China (82370737 to L.M., and 82100775 to L.L.), National Key R&D Program of China (2020YFC2005000 to L.M.), Sichuan Science/Technology Program (2022YFS0327 to L.L. and 2022YFS0589 to L.M.), and the 1.3.5 project for disciplines of excellence from West China Hospital of Sichuan University (ZYGD2023015 to P.F.).

## Author contributions

L.L. and L.M. designed the study. L.L., T.X., Y.W., S.T., and F.G. conducted the experiments. Y.W., J.G., and J.L. performed the bioinformatic analyzes. L.M., L.L., T.X., and Y.W. analyzed the data. L.L. and L.M. wrote the manuscript. L.M., H.F., and P.F. revised the language of the manuscript. All authors helped to interpret the results and approved the final version of the manuscript.

## Competing interests

The authors declare no competing interests.
