## [Peer Review File · Nature Communications]

Inhibition of ACSS2-mediated crotonylation alleviates kidney fibrosis via IL-1 β dependent macrophage activation and tubular cell senescenceREVIEWER COMMENTS

Reviewer #1 (Remarks to the Author):

In this manuscript, Li et al. have presented an intriguing study investigating the role of ACSS2-mediated H3K9cr in kidney fibrosis. The authors have explored the functional implications of H3K9cr, specifically its effect on the regulation of IL-1 β expression. Furthermore, they have studied how IL-1 β facilitates macrophage activation and induces tubular cell senescence, ultimately contributing to the progression of kidney fibrosis. Although the study holds promise, there are several issues that need to be addressed before considering the publication of this manuscript.

Major concerns:

1. The authors have demonstrated that H3K9cr exhibits high expression in the renal tissues of various chronic kidney disease patients and mouse models. However, they have also observed that the expression of ACSS2 remains unchanged in the mouse kidneys of UUO model (Fig S5D, Fig S21A), or even markedly decreases in the FAN model (Fig S5E, Fig S21A). This finding raises questions regarding the relationship between ACSS2 and the observed increase in H3K9cr levels. Moreover, it appears contradictory to the results obtained from ACSS2 knockout or inhibitor experiments, which showed a delay in the fibrosis process. To address these discrepancies, it is crucial for the authors to conduct a comprehensive analysis of both mRNA and protein expression profiles of ACSS2 in mouse models and human kidney tissue samples. This investigation will provide valuable insights into the underlying mechanisms driving the significant upregulation of H3K9cr levels in kidney tissues from various mouse models and patients with different etiologies of the disease.

2. It is worth noting that in Fig S21A, the ACSS2 inhibitor S8588 dramatically reduces ACSS2 levels in normal mice, which contradicts the observed upregulation of H3K9cr levels following S8588 treatment (Fig 8B). This observation is also inconsistent with the authors' previous in vitro experiments and their hypothesis. This inconsistency raises questions about the mechanism by which S8588 regulates ACSS2, especially considering its established role in inhibiting ACSS2-mediated acetyl-CoA synthesis, as reported in previous studies [Comerford et al., Cell, 2014]. To address this issue, the authors should provide an explanation for the potential mechanism underlying the observed reduction in ACSS2 expression upon S8588 treatment, and reconcile it with the observed changes in H3K9cr levels. Furthermore, it is important to assess the in vivo kinetics and safety profile of S8588. Such evaluation is crucial not only for understanding the pharmacokinetics and potential side effects of S8588 but also for determining its suitability as a therapeutic agent in clinical applications.

3. In this study, the authors have identified tubular epithelial cells as the primary affected cells based on the expression profile of H3K9cr and the results of ACSS2 conditional knockout. However, it is important to note that the mechanistic analysis was primarily conducted using RNA-seq and ChIP-seq on whole kidney tissues. During the progression of tubular cell injury and subsequent fibrosis induced by UUO or FAN, there are significant changes in the composition of renal cells, including a substantial infiltration of inflammatory cells. Therefore, it is crucial for the authors to validate the expression of key genes and the levels of H3K9cr in these genes in isolated tubular epithelial cells, at least through qPCR or ChIP-qPCR experiments. This will provide more specific insights into the expression patterns and histone modifications within the relevant cell type.

4. The authors observed a global increase in H3K9cr levels during fibrosis. To understand the mechanism underlying the selective regulation of gene expression resulting from changes in H3K9cr, the authors should investigate whether the regions exhibiting differential H3K9cr enrichment display specific binding of transcription factors. Exploring the establishment of specific transcription factor regulatory networks associated with H3K9cr can provide valuable insights into achieving gene expression specificity.

5. In addition to their impact on gene expression, histone acylation modifications also play important roles in maintaining genome stability. In the UUO or FAN-induced models, tubular epithelial cells are primarily affected, and changes in genome stability are key factors in the cells' response to the aforementioned injury. To investigate whether H3K9cr influences the damage response of renal tubular epithelial cells by affecting genome stability, the authors should examine DNA damage markers (eg. γ -H2AX) to assess the impact of altering ACSS2 expression levels or using ACSS2 inhibitors on the DNA damage response in both in vitro and in vivo injury models.

6. The authors proposed that the H3K9cr-mediated upregulation of IL-1 β in tubular epithelial cells facilitates the interaction between different cell types, including macrophage activation and tubular cell senescence. However, the aforementioned experiments were conducted using in vitro cell models and may not fully simulate the in vivo conditions. Importantly, during the process of injury, tubular epithelial cells exhibit diverse fate outcomes. Recent studies employing single-cell sequencing in mouse and human kidney injury and fibrosis have identified several novel tubular cell types responsible for injury response and renal fibrosis [Kuppe et al, *Nature*,2021; Li et al, *Cell Metab*,2022; Doke et al, *Nat Immunol*,2022; Balzer et al, *Nat Commun*,2022; McDaniels et al, *Kidney Int*,2023]. The authors should reanalyze the aforementioned single-cell data to examine the expression profile of ACSS2 and IL-1 β in different cell populations. Additionally, they should investigate the correlation between ACSS2 and IL-1 β expression and determine whether IL-1 β can mediate the observed interplay between different cell types in fibrotic disease models.

Minor concerns:

1. Quantitative statistical analysis is required for Fig1D, 1E, 2D, 3E, 7D, 7F, and 8D to accurately assess the significance of these immunoblotting results. In Fig 3C, the protein levels of H3K9cr in the UUO-CKO group were nearly undetectable, which is not consistent with the subsequent quantitative statistical findings.

2. The units for the vertical axis of all the qPCR analysis need to be specified.

3. Please note that there are inappropriate labels in some figures. In Fig 4A, "sequence" should be replaced with "sequencing". Please provide a clear description of what the three different colored lines (1,2, and 3) represent in Fig 4B. Please verify if the "+" and "-" symbols in Fig 6C and 6D have been incorrectly labeled. Additionally, it is recommended to replace "KO" with "ACSS2-/-" for better readability.

4. If the second row of these staining images in Fig1A, 3A, 3D, 5B, 7B, 7E, 8C and 8F represents an enlarged display of the first row, it is recommended to enclose the enlarged area. If they are not enlarged displays, it is advisable to clearly label each row image to indicate their respective content.

5. In Fig 4C, WT_C1_cr and WT_C2_cr should be compared with WT_UUO1_cr and WT_UUO2_cr. The same applies to the other ac groups. Furthermore, is the overall level of H3K9ac in the UUO group lower than that in the WT group?

6. Typically, the signals in RNA-seq are observed in the exon regions. However, the results shown in Fig 4F and Fig S10 do not appear to follow this pattern. Please verify if the gene locus positions have been incorrectly labeled.

7. As shown in Fig 5D, not all the tubular epithelial cells exhibit high expression of IL-1 β . To ascertain whether the cells expressing IL-1 β coincide with those exhibiting upregulated H3K9cr, it is suggested to perform immunofluorescence co-staining of IL-1 β and H3K9cr. Additionally, performing co-staining of IL-1 β with a macrophage marker would provide supporting evidence to confirm macrophage activation in vivo.

8. Please verify the correlation between the expression level of IL-1 β and the degree of fibrosis in patient tissues.

9. Please ensure that the uploaded images are of high resolution, as some of them appear blurry when magnified.

Reviewer #2 (Remarks to the Author):

Chronic kidney disease (CKD) is a global health concern characterized by kidney dysfunction, inflammation, and fibrosis. Current treatments for CKD are limited, necessitating the exploration of novel drug targets to arrest its progression. Previous studies implicated lysine crotonylation in acute kidney injury (AKI), however the role of histone crotonylation in CKD and fibrosis is not well understood. This manuscript aims to investigate the function of H3K9cr in kidney fibrosis and identify potential drug targets to attenuate kidney fibrosis.

The authors examined kidney samples from CKD patients and found that H3K9cr levels were significantly higher in patients with CKD compared to control. Mouse models of kidney fibrosis, including unilateral ureteric obstruction (UUO) and folic acid nephropathy (FAN), were used to confirm the association between H3K9cr and fibrosis. The expression of H3K9cr was increased in fibrotic kidneys, while H3K9ac (acetylation) levels remained stable. Specific histone lysine modifications, including H3K14cr, H3K18cr and H3K27cr, were also found to be increased in fibrotic kidneys. Importantly, the authors showed that genetic deletion and pharmacological inhibition of the enzyme Acetyl-CoA Synthetase 2 (ACSS2), which is responsible, at least in part, for the production of crotonyl-CoA to be used for histone crotonylation, decreased H3K9cr levels and alleviated kidney fibrosis in mice. Moreover, they showed that ACSS2-mediated H3K9cr was mainly present in tubular epithelial cells (TECs) of fibrotic kidneys. Tubular-specific deletion of ACSS2 delayed the progression of kidney fibrosis and reduced H3K9cr levels. Chromatin immunoprecipitation sequencing (ChIP-seq) and RNA sequencing (RNA-seq) analyses identified genes associated with H3K9cr and H3K9ac in fibrotic kidneys. The results indicated that H3K9cr can activate gene transcription and influence cytokine production and cytokine-cytokine receptor interactions. Indeed, they showed that H3K9 crotonylation promotes IL-1 β production in both kidney cells and fibrotic kidneys. Also, H3K9cr-induced IL-1 β promotes the process of tubular cell senescence both in vitro and in fibrotic kidneys. Altogether, the authors suggest that targeting the enzyme ACSS2 could be a potential therapeutic strategy to slow the progression of fibrotic kidney disease.

While the change in H3K9cr level and its role in modulating the expression of IL-1 β in kidney fibrosis are very interesting, several claims of this study are not supported by the presented data. Most importantly, the authors linked the ACSS2-dependent changes in H3K9cr to the production of IL-1 β , which subsequently promotes kidney fibrosis. However, they don't provide evidence supporting this conclusion. It is plausible to assume that pharmacological inhibition or knockout of ACSS2 enzyme, which is required for the synthesis of crotonyl-CoA, affects the crotonylation levels of not only H3K9cr but also other crotonylated lysine residues of histone and non-histone proteins. Therefore, the contribution of the changes in H3K9cr to the production of IL-1 β and kidney fibrosis remains correlative. In fact, the crotonylation of other lysine residues may also affect the expression of IL-1 β . In this regard, the authors claimed that the decrease in H3K9cr levels following ACSS2 inhibition leads to reduction in the expression of IL-1 β that partly contributes to kidney fibrosis. However, to prove this key point, the authors have to do some rescue experiments. For example, express IL-1 β in cells treated with ACSS2 inhibitor and show that it rescues the fibrotic phenotype.

The following are my additional concerns:

1. Figure 1C, the authors use Pan-Kcr antibody, therefore the increase in Kcr in UUO might be due to increase in the crotonylation of non-histone proteins. The author should repeat this using H3K9cr antibody.
2. Figure 1D, The intensity of the bands should be quantified and normalized to the loading control.

The same should be applied for all the western blots of this manuscript. Also, western using Pan-Kcr antibody should reveal several bands corresponding to the crotonylated 4 core histones. Hence, it would be beneficial to present the untrimmed blot, as it would display the extent of lysine crotonylation in histones other than H3 in both control and CKD samples.

3. Figure 1E, the increase in H3K9cr is very mild and not clear if it is significant.

4. Figure S2, the author showed nice increase in H3K14cr and H3K18cr in UUO and FAN samples, when compared to control. Is the crotonylation of H3K14cr and H3K18cr involved in regulating IL-1 β expression and kidney fibrosis?

5. Figure S3A and S4B: the authors showed that ACSS2 overexpression (Fig. S3A) reduces the levels of H3K9cr (Fig. S4B). However, it is not convincing. The authors should show this in the same blot and using the same samples.

6. The reason behind the absence of an effect on H3K9cr levels in control mice (Figure 2B) upon ACSS2 knockout is unclear. To the best of my knowledge, ACSS2 is functional in both control and UUO, thus we should expect an increase in H3K9cr even in control mice.

7. The authors should map all ASCC2 dependent alterations in Kcr residues in their model cell lines and find whether other residues behave like H3K9 in term of increase in Kcr but not Kac.

8. In page 12, the authors concluded that ACSS2 is a key regulator of H3K9cr-mediated IL-1b production. However, they didn't provide any evidence showing that the increase in IL-1B production is caused by the elevated levels of H3K9cr at the IL-1b promoter.

9. The conclusion that H3K9cr promoted cytokine production and regulated cytokine-cytokine receptor interaction in fibrotic kidneys lacks experimental evidence and is based solely on high-throughput experiments.

10. In figure 5D, there is a lack of correlation between the mRNA levels and the protein expression.

11. Is there any correlation between ASCC2 expression / H3K9cr and kidney diseases?

12. The authors used the term "H3K9cr expression" it should be changed to "H3K9cr levels"

13. Some citations are missing. For examples, the paragraph that starts at line 50 is missing references.

14. The text contains typos and grammatical errors that need to be revised.

Reviewer #3 (Remarks to the Author):

The authors conducted a functional study showing an increase of histone Kcr in fibrotic kidneys. Furthermore, the levels of Kcr (i.e. H3K9cr) were found to be reduced by higher levels of ACSS2 which in turn reduced IL1b expression and subsequently reduced macrophage activation and tubular cell senescence. These effects are finally attenuating kidney fibrosis. The paper is well written, the analyses are very comprehensive and sound, and the results generally support the conclusions.

My major concern is related to the causal link that the manipulation of histone Kcr by ACSS2 inhibition really results in a reduced kidney fibrosis state, which is needed to justify the claimed therapeutic potential. To underpin this relation, the authors presented several staining images (e.g. Fig. 3D, 2C, 7B) and blots (e.g. Fig. 1D, 1E, 2B, 3E, 5E, 7F). However, it is often difficult for me to see the differences between the fibrotic and reduced fibrotic tissue images, and the stated differences among groups of blots (taking into account that there is also variation in expression among controls). There should be a possibility to quantify these differences, or at least describe the specific changes in more detail (i.e. for the staining images).

Additional comments are:

- After reading the manuscript, it turned out that all experiments except the correlation of Histone Kcr expression with CKD was performed in mice or cell lines. This needs to be clearly stated in the manuscript, including the abstract, the methods and the figure legends.

- Basis characteristics of the patients are missing. Although the sample size could be guessed from Fig. 1B, how many males/females were included, what was the age range, proportion of CKD? Which formula was used to estimate the eGFR? Are the correlation results in this figure comparable after adjusting for common covariates (e.g. sex and age)?

- Almost all Figure descriptions contain many abbreviations, which are only explained in the main text. The Figure descriptions should be readable without having to go back to the main text to understand the abbreviations. Specifically, please describe the meaning of the abbreviations and of the arrows in Fig. 1A and Fig. S1A. Furthermore, I suggest to improve the description and/or labels of Fig. 1C and Fig. S9: only providing labels like WT_C1_cr is hard to follow.

- Pathway analysis: it was stated that after ACSS2 deletion in mice the cytokine pathways decreased (Fig. S10 B-E). What does it mean exactly? As far as I see, the GeneRatio and p-values increased, and Count decreased which is the case for all pathways listed in these figure panels. Furthermore, please explain what GeneRatio and Count stand for and how to interpret them (also for Fig. 4E).

- Description of Fig. 4B: I assume it should state "tertile of gene expression" instead of "quartile".

- Fig. S10 H: The figure description states qPCR analysis of H3K9Cr and H3K9ac ChIP products, but the labels show solely H3K9cr. Please check and clarify, which data is shown.

No: NCOMMS-23-24253

Title: Inhibition of ACSS2-mediated H3K9 crotonylation alleviates kidney fibrosis via IL-1 β -dependent macrophage activation and tubular cell senescence

Dear Editors and Reviewers,

We sincerely thank you for thoroughly examining our manuscript and providing helpful comments to guide our revision. We have tried our best to revise and improve our manuscript according to your kind comments point to point. Please find the following detailed responses to your comments and suggestions. Besides revised manuscript, supplementary data, response to reviewers, independent figures and raw data, we also upload reporting summary and editorial policy checklist in the system.

We sincerely hope that this revised manuscript has addressed all your comments and suggestions. We appreciated for reviewers' warm work earnestly, and hope that the correction will meet with approval.

Yours sincerely,

Prof. Dr. Liang Ma (on behalf of the authors)

REVIEWER COMMENTS

Reviewer #1 (Remarks to the Author):

In this manuscript, Li et al. have presented an intriguing study investigating the role of ACSS2-mediated H3K9cr in kidney fibrosis. The authors have explored the functional implications of H3K9cr, specifically its effect on the regulation of IL-1 β expression. Furthermore, they have studied how IL-1 β facilitates macrophage activation and induces tubular cell senescence, ultimately contributing to the progression of kidney fibrosis. Although the study holds promise, there are several issues that need to be addressed before considering the publication of this manuscript.

Major concerns:

1. The authors have demonstrated that H3K9cr exhibits high expression in the renal tissues of various chronic kidney disease patients and mouse models. However, they have also observed that the expression of ACSS2 remains unchanged in the mouse kidneys of UUO model (Fig S5D, Fig S21A), or even markedly decreases in the FAN model (Fig S5E, Fig S21A). This finding raises questions regarding the relationship between ACSS2 and the observed increase in H3K9cr levels. Moreover, it appears contradictory to the results obtained from ACSS2 knockout or inhibitor experiments, which showed a delay in the fibrosis process. To address these discrepancies, it is crucial for the authors to conduct a comprehensive analysis of both mRNA and protein expression profiles of ACSS2 in mouse models and human kidney tissue samples. This investigation will provide valuable insights into the underlying mechanisms driving the significant upregulation of H3K9cr levels in kidney tissues

from various mouse models and patients with different etiologies of the disease.

Response: Thank you for your helpful comments and suggestions. According to your suggestions, we firstly stained ACSS2 and H3K9cr in human kidney biopsy slides and found the increase of unclear ACSS2 in patients with kidney fibrosis (Fig S9A). However, no matter in UUO or FAN mice, the expression of ACSS2 from bulk kidneys was decreased as shown in western blot result (Fig S5D-E). Interestingly, genetic and pharmacological inhibition of ACSS2 in UUO or FAN mice could effectively alleviate kidney fibrosis.

In order to figure out this discrepancy to explain the underlying mechanisms, we checked the published single-cell transcriptomic data according to your suggestions. By analysis of Balzer et al, Nat Commun 2022 and Wu et al, JASN 2019, we found the decrease of ACSS2 in different segments of proximal tubule (PT) (Fig S6A), while the level of ACSS2 in PT-S3 was significantly increased in patients with kidney fibrosis from human single-cell data (Lake et al, Nature, 2023) (Fig S6B). After rechecking the IHC staining of ACSS2 in our kidney fibrotic mice, we confirmed that the increase of ACSS2 was mainly expressed in corticomedullary junction which showed strong spatial characteristics (Fig S6C). From our analysis, for cells with obvious spatial characteristics, the ability of single-cell transcriptomics to distinguish tubular cells is far inferior to spatial transcriptomics due to the issue of how much and where sample is taken from (Fig S6D-E). Mouse single-cell transcriptomics typically selects a piece of cortex for testing, while human kidney single-cell transcriptomics always uses intact cortical and medullary kidney penetrating tissue for subsequent

experiments. The differences in single-cell transcriptome results indicate that different sampling methods have an impact on the expression pattern of ACSS2, which suggested spatial transcriptomics might be appropriate methods for this study of ACSS2 location and expression. After reanalyzing the spatial transcriptomics data from Dixon et al, JASN 2021, we found that PT-S3 cells were those cells mainly expressed in corticomedullary junction (Fig S6F-G), where ACSS2 was obviously increased in those corticomedullary junction PT-S3 cells in fibrotic kidney (Fig S6H). By integrated analysis of spatial transcriptomics data and our experiments, the increase of ACSS2 is seemed to be in especially spatial tubular epithelial cells, while ACSS2 inhibition do exert protective roles against kidney fibrosis. Unfortunately, we do not conduct spatial transcriptomics in our tissue. Hoping we could have further analysis to explore the functions of these special cells with spatial characteristics in the future.

2. It is worth noting that in Fig S21A, the ACSS2 inhibitor S8588 dramatically reduces ACSS2 levels in normal mice, which contradicts the observed upregulation of H3K9cr levels following S8588 treatment (Fig 8B). This observation is also inconsistent with the authors' previous in vitro experiments and their hypothesis. This inconsistency raises questions about the mechanism by which S8588 regulates ACSS2, especially considering its established role in inhibiting ACSS2-mediated acetyl-CoA synthesis, as reported in previous studies [Comerford et al., Cell, 2014]. To address this issue, the authors should provide an explanation for the potential mechanism

underlying the observed reduction in ACSS2 expression upon S8588 treatment, and reconcile it with the observed changes in H3K9cr levels. Furthermore, it is important to assess the in vivo kinetics and safety profile of S8588. Such evaluation is crucial not only for understanding the pharmacokinetics and potential side effects of S8588 but also for determining its suitability as a therapeutic agent in clinical applications.

Response: Thank you so much for your comments and suggestions. For the first question, the results of Fig 8B showed that inhibition of ACSS2 could reduce the level of H3K9cr in kidneys of UUO mice. As the sample order in Fig 8B is not consistent as those in Fig S29A and S30A, we are sorry to make you misunderstand the results. We have marked UUO in red to help read this figure much more easily.

As to the second suggestion, according to [Comerford et al., Cell, 2014], ACSS2 inhibitor named VY-3-249, a small molecule quinoxaline with an IC₅₀ of ~0.6 μM in biochemical assays, and ~5 μM in its ability to inhibit cellular [¹⁴C] acetate uptake into both lipids and histones, emerged as one of the most favorable inhibitors. Unfortunately, the detail mechanism of how could VY-3-249 inhibit ACSS2 is rarely mentioned. The previous study found that ACSS2 regulated H3K9cr level by mediating the production of crotonyl-CoA (Sabari, B. R. *et al. Mol Cell*, 2015). So, we supposed that VY-3-249 might inhibit ACSS2 expression and thus to reduce crotonyl-CoA, the resources of H3K9cr, which finally suppress H3K9cr level.

Finally, according to your suggestions, we have added the experiments to assess the pharmacokinetics and safety profile of ACSS2 inhibitor VY-3-249. The pharmacokinetic results of VY-3-249 have been added in supplemental data Table S3.

Methods for pharmacokinetic have been added in Material and Methods in supplementary file. The safety of VY-3-249 was assessed as well. There are no differences of kidney and liver function between control and VY-3-249 administration group (Fig S28A). The tissue staining revealed that no damage of heart, lung, liver and spleen in VY-3-249 administration group with appropriate pharmacokinetic parameters for potential clinical applications (Table S3 and Fig S28A-B).

3. In this study, the authors have identified tubular epithelial cells as the primary affected cells based on the expression profile of H3K9cr and the results of ACSS2 conditional knockout. However, it is important to note that the mechanistic analysis was primarily conducted using RNA-seq and ChIP-seq on whole kidney tissues. During the progression of tubular cell injury and subsequent fibrosis induced by UUU or FAN, there are significant changes in the composition of renal cells, including a substantial infiltration of inflammatory cells. Therefore, it is crucial for the authors to validate the expression of key genes and the levels of H3K9cr in these genes in isolated tubular epithelial cells, at least through qPCR or ChIP-qPCR experiments. This will provide more specific insights into the expression patterns and histone modifications within the relevant cell type.

Response: Thank you so much for these suggestions which is really important for our study. This is true that inflammatory cells infiltrated into injured kidneys during fibrosis process and changed the composition of renal cells. In order to validate the relationship between ACSS2 and H3K9cr in tubular epithelial cells, we have treated

both tubular epithelial TCMK-1 cells and human embryo kidney epithelial-like HEK-293T cells using ACSS2 overexpressing plasmids, and found that ACSS2 overexpression could rise H3K9cr expression and thus to influence Il1b expression (Fig S3A-3B and Fig S4B).

In addition, according to your suggestions, we collected the primary tubular epithelial cells from WT and ACSS2^{-/-} mice and treated them with TGFβ1 to further check the relationship between ACSS2 and H3K9cr-Il1b axis. The results are consistent with the previous findings that link the relationship between ACSS2 and H3K9cr in tubular epithelial cells (Fig S16K-M). Besides qPCR experiments, the increase in Il1b and Il1r1 enrichment of H3K9cr following transfection of the ACSS2 plasmid in HEK-293T cells was further confirmed by ChIP-qPCR (Fig. S13E). Thank you again for your helpful suggestions again.

4. The authors observed a global increase in H3K9cr levels during fibrosis. To understand the mechanism underlying the selective regulation of gene expression resulting from changes in H3K9cr, the authors should investigate whether the regions exhibiting differential H3K9cr enrichment display specific binding of transcription factors. Exploring the establishment of specific transcription factor regulatory networks associated with H3K9cr can provide valuable insights into achieving gene expression specificity.

Response: Thanks a million for these suggestions. We have supplemented the motifs of H3K9cr ChIP-seq both in control and UUO mice. By analysis, the top five motifs

of HNF1b, COUP-TFII, ERRg, HNF4a, HNF1 were shown in control kidney tissue, while ERG, ETV2, ETS1, GABPA, Fli1 were shown in kidneys of UUO mice (Fig. S13F). As known, transcription factors are crucial to regulate gene expression, and we are planning to explore their potential function related to H3K9cr in kidney fibrosis, hoping to get more data soon.

5. In addition to their impact on gene expression, histone acylation modifications also play important roles in maintaining genome stability. In the UUO or FAN-induced models, tubular epithelial cells are primarily affected, and changes in genome stability are key factors in the cells' response to the aforementioned injury. To investigate whether H3K9cr influences the damage response of renal tubular epithelial cells by affecting genome stability, the authors should examine DNA damage markers to assess the impact of altering ACSS2 expression levels or using ACSS2 inhibitors on the DNA damage response in both in vitro and in vivo injury models.

Response: Thanks for these helpful suggestions. Accordingly, we have stained kidney and cell samples (the primary tubular epithelial cells and HEK-293T) using γ H2AX to assess the content of DNA damage in Fig. S24A-B. No matter in UUO mice or TGF β 1-stimulated cells, the DNA damage is increased, while it could be suppressed by genetic deletion and inhibitor of ACSS2, suggesting that ACSS2-mediated H3K9cr play important roles in maintaining genome stability in kidney fibrosis.

6. The authors proposed that the H3K9cr-mediated upregulation of IL-1 β in tubular

epithelial cells facilitates the interaction between different cell types, including macrophage activation and tubular cell senescence. However, the aforementioned experiments were conducted using *in vitro* cell models and may not fully simulate the *in vivo* conditions. Importantly, during the process of injury, tubular epithelial cells exhibit diverse fate outcomes. Recent studies employing single-cell sequencing in mouse and human kidney injury and fibrosis have identified several novel tubular cell types responsible for injury response and renal fibrosis [Kuppe et al, Nature, 2021; Li et al, Cell Metab, 2022; Doke et al, Nat Immunol, 2022; Balzer et al, Nat Commun, 2022; McDaniels et al, Kidney Int, 2023]. The authors should reanalyze the aforementioned single-cell data to examine the expression profile of ACSS2 and IL-1 β in different cell populations. Additionally, they should investigate the correlation between ACSS2 and IL-1 β expression and determine whether IL-1 β can mediate the observed interplay between different cell types in fibrotic disease models.

Response: Thanks for your helpful suggestions. Firstly, according to your proposal, we conducted *in vivo* study to confirm the roles of H3K9cr-mediated upregulation of IL-1 β in regulating macrophage activation and senescence of tubular epithelial cells. The injection of anti-IL-1 β antibody could inhibit macrophage activation and senescence of tubular epithelial cells to delay kidney fibrosis as shown in Fig 7. Besides, we treated mice with ACSS2 inhibitor to decrease ACSS2 expression followed by stimulation by IL-1 β . We found that the inhibition of macrophage activation and senescence of tubular epithelial cells by suppression of ACSS2 could be reversed by IL-1 β (Fig S26-27). Both of the studies could help to explain the roles

of H3K9cr-mediated upregulation of IL-1 β in regulating macrophage activation and senescence of tubular epithelial cells in fibrotic kidneys.

Second, with your suggestion, we analyzed the published single-cell data and spatial transcriptomics data as what we mentioned in question 1. As well, from Katalin's single-cell analysis also mentioned that inflammatory signaling connections were active in maladaptive tubular epithelial cells (TECs) and myeloid cells, as well as among epithelial cells in fibrotic kidneys; The IL-1 β is proved to be the key hub gene (Balzer et al, Nat Commun, 2022). From the spatial transcriptomics data, the correlation between ACSS2 and IL-1 β expression in tubular epithelial cells was shown in Fig S14C. The positional relationships of different cells in the spatial transcriptome are also shown in Fig S6F and Figure S19A. According to spatial transcriptome analysis, the TECs with increased ACSS2 were close to macrophage; thus, it is possible that H3K9cr-mediated IL-1 β triggers macrophage activation to promote kidney fibrosis progression.

Minor concerns:

1. Quantitative statistical analysis is required for Fig1D, 1E, 2D, 3E, 7D, 7F, and 8D to accurately assess the significance of these immunoblotting results. In Fig 3C, the protein levels of H3K9cr in the UUO-CKO group were nearly undetectable, which is not consistent with the subsequent quantitative statistical findings.

Response: Thanks for these useful suggestions. We have quantified these blots and supplement them in Fig S1D, S1F, S7C, and S29G, respectively. For Fig3D, 7D and

7F, the quantifications of these blots have been already in Fig S10B and Fig S25, with no change. As to the second question, as the density of blot in UUO-WT group is too strong, other blots, especially those in UUO-CKO group looks mild. In addition, the amount of H3 is less in UUO-CKO group. So, the quantification of bolts in UUO-CKO group is appropriate and expected. In order to avoid misunderstand of readers, we have redone the western blot experiments and incubated with first antibody for longer time to get a better blot. We have changed the blots in Fig 3C. Thank you again for your careful reading that do helps to make the paper better.

2. The units for the vertical axis of all the qPCR analysis need to be specified.

Response: Thanks for your careful suggestion. The vertical axis for these qPCR analyses should be as fold change of genes. We have modified all these panel in our revised manuscript.

3. Please note that there are inappropriate labels in some figures. In Fig 4A, "sequence" should be replaced with "sequencing". Please provide a clear description of what the three different colored lines (1, 2, and 3) represent in Fig 4B. Please verify if the "+" and "-" symbols in Fig 6C and 6D have been incorrectly labeled. Additionally, it is recommended to replace "KO" with "ACSS2-/-" for better readability.

Response: Thanks for your careful suggestion. We have replaced "sequence" with "sequencing" in Fig 4A. For second one, in order to make it clear, we have changed it

to “(B) All genes were split into three equal groups based on their expression levels, calculated using RNA-seq. The mean H3K9cr and H3K9ac ChIP signals are shown for each tertile of gene expression, and are shown using different colored lines (the red line is ChIP signals of the gene with the highest expression, the green line is ChIP signals of the gene with the intermediate expression, the blue line ChIP signals of the gene are with the lowest expression) in both control and UUO mice”. Thirdly, we have corrected the mislabel of "+" and "-" symbols in Fig 6C and 6D. Finally, we have replaced all “KO” with "ACSS2^{-/-}" in figures as you suggestion. And we have checked and corrected these problems in our revised manuscript. These really helps the paper better!

4. If the second row of these staining images in Fig1A, 3A, 3D, 5B, 7B, 7E, 8C and 8F represents an enlarged display of the first row, it is recommended to enclose the enlarged area. If they are not enlarged displays, it is advisable to clearly label each row image to indicate their respective content

Response: Thanks for your careful suggestion. As for these images of Fig1A, 3A, 3D, 5B, 7B, 7E, and 8F, we have enclosed the enlarged area. The Fig 8C is Masson and Sirius red staining in the same magnification but not on the same slide, so we do not enclose the enlarged area in this panel.

5. In Fig 4C, WT_C1_cr and WT_C2_cr should be compared with WT_UUO1_cr and WT_UUO2_cr. The same applies to the other ac groups. Furthermore, is the overall

level of H3K9ac in the UUO group lower than that in the WT group?

Response: Thanks for your careful suggestion. The comparison between WT_C_cr with WT_UUO_cr has been supplemented in Fig S12A. From our ChIP seq data, the overall level of H3K9ac in the UUO group is lower than that in the WT group.

6. Typically, the signals in RNA-seq are observed in the exon regions. However, the results shown in Fig 4F and Fig S10 do not appear to follow this pattern. Please verify if the gene locus positions have been incorrectly labeled.

Response: Thanks for your careful suggestion. We have checked the data again. We have modified Fig 4F and Fig S13D by decreasing the track range. The signals in RNA-seq are in the exon regions as you suggested. Previous signals from non-exon regions might be some non-coding RNA signals. Thank you again for your mindful reading.

7. As shown in Fig 5D, not all the tubular epithelial cells exhibit high expression of IL-1 β . To ascertain whether the cells expressing IL-1 β coincide with those exhibiting upregulated H3K9cr, it is suggested to perform immunofluorescence co-staining of IL-1 β and H3K9cr. Additionally, performing co-staining of IL-1 β with a macrophage marker would provide supporting evidence to confirm macrophage activation *in vivo*.

Response: Thanks for your careful suggestion. We have conducted immunofluorescence co-staining of IL-1 β and H3K9cr in fibrotic kidneys (Fig 5F and Fig S14D). We found that those tubular epithelial cells which highly expressed

H3K9cr (yellow) could be co-stained with IL-1 β (red). Additionally, we also co-staining of IL-1 β with macrophage marker (F4/80) and a M1 macrophage marker (iNOS) in Fig S20E. Interestingly, IL-1 β is mainly expressed in tubular epithelial cells instead of macrophage, which has been confirmed that just a few cells were co-stained with IL-1 β and macrophage markers. Instead, when we co-stained macrophage marker with IL-1 β receptor, it could be found the colocalization of IL-1 β receptor (Il1r1) and macrophage markers. It is strongly supported our hypothesis that ACSS2-mediated H3K9cr induced the secretion of IL-1 β in tubular epithelial cells and thus to stimulate activation of macrophage by targeting IL-1 β receptor (Fig S20F).

8. Please verify the correlation between the expression level of IL-1 β and the degree of fibrosis in patient tissues.

Response: Thanks for your careful suggestion. We have stained IL-1 β in human kidney biopsy slides and we found the expression of IL-1 β is negatively correlated with eGFR as shown in Fig S14A-B.

9. Please ensure that the uploaded images are of high resolution, as some of them appear blurry when magnified.

Response: Thanks for your careful suggestion. As the upload system suggested us to insert the figures in manuscript which scarified some parts of the picture quality. In this revision process, we will upload the high-resolution figures for reviewing.

Reviewer #2 (Remarks to the Author):

Chronic kidney disease (CKD) is a global health concern characterized by kidney dysfunction, inflammation, and fibrosis. Current treatments for CKD are limited, necessitating the exploration of novel drug targets to arrest its progression. Previous studies implicated lysine crotonylation in acute kidney injury (AKI), however the role of histone crotonylation in CKD and fibrosis is not well understood. This manuscript aims to investigate the function of H3K9cr in kidney fibrosis and identify potential drug targets to attenuate kidney fibrosis. The authors examined kidney samples from CKD patients and found that H3K9cr levels were significantly higher in patients with CKD compared to control. Mouse models of kidney fibrosis, including unilateral ureteric obstruction (UUO) and folic acid nephropathy (FAN), were used to confirm the association between H3K9cr and fibrosis. The expression of H3K9cr was increased in fibrotic kidneys, while H3K9ac (acetylation) levels remained stable. Specific histone lysine modifications, including H3K14cr, H3K18cr and H3K27cr, were also found to be increased in fibrotic kidneys. Importantly, the authors showed that genetic deletion and pharmacological inhibition of the enzyme Acetyl-CoA Synthetase 2 (ACSS2), which is responsible, at least in part, for the production of crotonyl-CoA to be used for histone crotonylation, decreased H3K9cr levels and alleviated kidney fibrosis in mice. Moreover, they showed that ACSS2-mediated H3K9cr was mainly present in tubular epithelial cells (TECs) of fibrotic kidneys. Tubular-specific deletion of ACSS2 delayed the progression of kidney fibrosis and reduced H3K9cr levels. Chromatin immunoprecipitation sequencing (ChIP-seq) and

RNA sequencing (RNA-seq) analyses identified genes associated with H3K9cr and H3K9ac in fibrotic kidneys. The results indicated that H3K9cr can activate gene transcription and influence cytokine production and cytokine-cytokine receptor interactions. Indeed, they showed that H3K9 crotonylation promotes IL-1 β production in both kidney cells and fibrotic kidneys. Also, H3K9cr-induced IL-1 β promotes the process of tubular cell senescence both in vitro and in fibrotic kidneys. Altogether, the authors suggest that targeting the enzyme ACSS2 could be a potential therapeutic strategy to slow the progression of fibrotic kidney disease.

While the change in H3K9cr level and its role in modulating the expression of IL-1 β in kidney fibrosis are very interesting, several claims of this study are not supported by the presented data. Most importantly, the authors linked the ACSS2-dependent changes in H3K9cr to the production of IL-1 β , which subsequently promotes kidney fibrosis. However, they don't provide evidence supporting this conclusion. It is plausible to assume that pharmacological inhibition or knockout of ACSS2 enzyme, which is required for the synthesis of crotonyl-CoA, affects the crotonylation levels of not only H3K9cr but also other crotonylated lysine residues of histone and non-histone proteins. Therefore, the contribution of the changes in H3K9cr to the production of IL-1 β and kidney fibrosis remains correlative. In fact, the crotonylation of other lysine residues may also affect the expression of IL-1 β . In this regard, the authors claimed that the decrease in H3K9cr levels following ACSS2 inhibition leads to reduction in the expression of IL-1 β that partly contributes to kidney fibrosis. However, to prove this key point, the authors have to do some rescue experiments.

For example, express IL-1 β in cells treated with ACSS2 inhibitor and show that it rescues the fibrotic phenotype.

Response: Thanks for your careful and helpful suggestion. **Firstly**, according to your comments, we have supplemented the changes of crotonylation and acetylation levels in other histone lysine in ACSS2^{-/-} mice in Fig S7A, S7B, S8C and S8D. Here, gene knockout of ACSS2 could downregulate the levels of H3K18cr and H3K27ac, while have no influence on H3K18ac and H3K27cr levels, which suggested the influence of ACSS2 on histone lysine crotonylation and acetylation might not only be limited to H3K9 site. However, according to our data from kidney fibrotic mice, the increase of H3K9cr level is much more obvious than that of H3K9ac, which is not the same pattern as other lysine (Fig 1E and S2H). So, with interest, we first studied the H3K9 site and found that ACSS2 could regulate H3K9cr-mediated renal fibrosis. In the meantime, we did agree with the role of other lysine modification. The question of which site and which modification play more important roles is a huge one. We will gradually explore the role of crotonylation/acetylation modifications at various sites in renal fibrosis. I hope that we could conduct joint analysis and discover the most important site modifications in future.

Secondly, as to rescue experiments, we have already conducted the *in vitro* studies as shown in Fig 6. We overexpressed tubular epithelial cells with ACSS2 plasmids and collected the supernatants. The macrophage activation and tubular epithelial cell senescence triggered by these supernatants could be reversed by Anti-IL-1 β Ab. According to your suggestions, we have conducted the rescue experiments *in vivo*. We

treated these UUO mice with ACSS2 inhibitor followed by IL-1 β stimulation (Fig S26-27). We found that the injection of ACSS2 inhibitor could alleviate the fibrosis, cellular senescence, and inflammation of UUO kidneys, while IL-1 β could eliminate these benefits. Thank you again for all your mindful suggestions that do helps the study better!

The following are my additional concerns:

1. Figure 1C, the authors use Pan-Kcr antibody, therefore the increase in Kcr in UUO might be due to increase in the crotonylation of non-histone proteins. The author should repeat this using H3K9cr antibody.

Response: Thanks for your careful suggestion. Accordingly, we have repeated these experiments using H3K9cr antibody and replaced these pictures in Figure 1C. The results of Pan-Kcr antibody in UUO has been moved to supplemental Fig S1D.

2. Figure 1D, the intensity of the bands should be quantified and normalized to the loading control. The same should be applied for all the western blots of this manuscript. Also, western using Pan-Kcr antibody should reveal several bands corresponding to the crotonylated 4 core histones. Hence, it would be beneficial to present the untrimmed blot, as it would display the extent of lysine crotonylation in histones other than H3 in both control and CKD samples.

Response: Thanks for your careful suggestion. In the revised manuscript, we have repeated the western blot for pan-Kcr staining, together with Coomassie Blue staining

and reload untrimmed blots in Fig S1D-E. There is a slight H4 histone line under the H3 histone after Pan-Kcr antibody incubation, which shows the similar pattern as what we found in H3 histone. The other qualification results of these western blots in Figures have also been applied in supplementary figures.

3. Figure 1E, the increase in H3K9cr is very mild and not clear if it is significant.

Response: Thanks for your careful suggestion. We have repeated the experiments of fibrotic kidneys and replace the blot of H3K9cr in Fig 1E.

4. Figure S2, the author showed nice increase in H3K14cr and H3K18cr in UUO and FAN samples, when compared to control. Is the crotonylation of H3K14cr and H3K18cr involved in regulating IL-1 β expression and kidney fibrosis?

Response: Thanks for your comments, and this is a very interesting question to our study. In fact, we could confirm that ACSS2-mediated H3K9 crotonylation alleviated kidney fibrosis via IL-1 β -dependent mechanism. And the crotonylation at H3K14 and H3K18 site is also involved in kidney fibrosis, but whose function via regulating IL-1 β expression is more important has not been explored in the study. As known, precise regulation of crotonylation is systematic and complicated, so, this is actually a big topic to explore the functions of different site of crotonylated modifications. We apologize for not being able to complete all the content in this study, but we will gradually analyze the role of crotonylation at each site, and look forward to one day organizing a graph of the relationship and functions of histone crotonylation at each

site in renal fibrosis to discover potential drug targets.

5. Figure S3A and S4B: the authors showed that ACSS2 overexpression (Fig. S3A) reduces the levels of H3K9cr (Fig. S4B). However, it is not convincing. The authors should show this in the same blot and using the same samples.

Response: Thanks for your careful suggestion. In the Figure S3A and S4B of our study, the overexpression of ACSS2 increase the level of H3K9cr. We used same samples for these blots and we incubated these antibodies in the same bolt and added it in raw data. The data of Fig S3 showed the influences of different treatment, while Fig S4 revealed the changes of H3K9cr and H3K9ac. In order to make the paper more logic, we are not supposed to combine these blots into one figure. We still feel great thanks for your suggestion.

6. The reason behind the absence of an effect on H3K9cr levels in control mice (Figure 2B) upon ACSS2 knockout is unclear. To the best of my knowledge, ACSS2 is functional in both control and UUO, thus we should expect an increase in H3K9cr even in control mice.

Response: Thanks for this interesting question. It is helpful to us. The decrease of H3K9cr after ACSS2 knockout in control mice is not actually significant, but ACSS2 have more influences on H3K9cr level in the fibrotic kidneys (Figure 2B). We speculate that in the physiological state, the body has a balance of other genes (perhaps ACSS1, HDAC, SIRT, CBP/P300, etc) which could compensate the deletion

effects from ACSS2. However, in fibrotic state, the balance has been broken by tissue damage or sustained inflammation, which augmented the roles of ACSS2-mediated H3K9cr. The complete regulatory mechanism of H3K9cr still needs further exploration in order to better understand why the knockout of ACSS2 in the control kidney does not affect the level of H3K9cr, while genetic deletion of ACSS2 exert a significant impact in the kidney fibrosis model. Thank you again for the big new topic we could explore in the future.

7. The authors should map all ASCC2 dependent alterations in Kcr residues in their model cell lines and find whether other residues behave like H3K9 in term of increase in Kcr but not Kac.

Response: Thanks for this useful suggestion. We have supplied the changes of crotonylation and acetylation levels in other histone lysine in ACSS2^{-/-} mice in Fig S7A, S7B, S8C, and S8D. The knockout of ACSS2 could decrease the levels of H3K18cr and H3K27ac, while have no influence on H3K18ac and H3K27cr, which suggested the influence of ACSS2 on crotonylation and acetylation of histone lysine might be complicated. However, as there are several residues which made us impossible to check changes of all residues by western blot. The LC-MS/MS analysis should be better choice in the future. However, with the limitation of antibody production, our study cannot reach that high level now. In this study, according to our data in kidney fibrosis mice, the increase of H3K9cr level is much more obvious than that of H3K9ac, which is not the same pattern as other lysine (Fig S1E and S2H-K).

So, we first studied this H3K9 site and found that ACSS2 could regulate H3K9cr-mediated renal fibrosis. But we did not negate the role of other lysine. The question of which site and which modification have a more important role is a huge one. We will gradually improve the role of crotonylation and acetylation modifications at various sites in renal fibrosis. I hope that one day we can conduct joint analysis and discover the most important site modifications.

8. In page 12, the authors concluded that ACSS2 is a key regulator of H3K9cr-mediated IL-1b production. However, they didn't provide any evidence showing that the increase in IL-1B production is caused by the elevated levels of H3K9cr at the IL-1b promoter.

Response: Thank you for your comments. We are sorry for the misunderstanding of these results. We have added several experiments and we are willing to briefly conclude the results here. Firstly, the ChIP-seq data revealed the enrichment of H3K9cr on IL-1b promoter, which will be augmented in UUO fibrotic models (Fig 4F). The ChIP-qPCR have also confirmed that overexpression of ACSS2 could increase the enrichment of H3K9cr on IL-1b and IL1r1 in epithelial cells (Fig S13E). Next, we treated tubular epithelial cells using different ways to overexpress, maintain or suppress H3K9cr level to check the correlation between H3K9cr and IL-1b levels (Fig 5A, Fig S14-18). In addition, we added the experiment to confirm the colocalization of H3K9cr and IL-1b in mice (Fig 5F and S14D), and also found that knock out of ACSS2 *in vivo* will also suppress IL-1b expression (Fig 5D-E and S15C-D). Finally,

we conducted rescue studies *in vitro* and *in vivo* according to your suggestion. In cells, we overexpressed tubular epithelial cells with ACSS2 plasmids and collected the supernatants. The activation of macrophage and senescence of tubular epithelial cells caused by these supernatants could be reversed by Anti-IL-1 β Ab (Fig 6). We have also treated UUO mice with ACSS2 inhibitor followed by IL-1 β stimulation (Fig S26-27). We found that injecting of ACSS2 inhibitor could alleviate the fibrosis, cellular aging, and inflammation of UUO kidneys while IL-1 β could eliminate these benefits. All of these data above supported that ACSS2 was a key regulator of H3K9cr-mediated IL-1 β production.

9. The conclusion that H3K9cr promoted cytokine production and regulated cytokine-cytokine receptor interaction in fibrotic kidneys lacks experimental evidence and is based solely on high-throughput experiments.

Response: Thanks for your comments which give us opportunities to explain here. Firstly, based on ChIP-seq data, we found that H3K9cr might promote cytokine-mediated signaling pathway and regulated cytokine-cytokine receptor interaction in fibrotic kidneys (Fig 4E, S12C-D, and S13A-C). Combined with previous study, Katalin's single-cell analysis, IL-1 β was proved to be the important hub gene came into our consideration (Balzer et al, Nat Commun,2022). Next, in *in vitro* and *in vivo* studies, we found that H3K9cr-mediated IL-1 β could stimulate cellular senescence of tubular epithelial cells and thus to promote the expression of senescence-associated-secreted protein (which included several inflammatory cytokines). As well, H3K9cr-

mediated IL-1 β could activate M1 macrophage, which is a proinflammatory types of macrophages (Fig 6, S19-21). According to your suggestions, we have conducted the rescue experiments. We treated UUO mice with ACSS2 inhibitor to inhibit H3K9cr level followed by IL1 β stimulation (Fig S26-27). We found that injecting of ACSS2 inhibitor could alleviate the fibrosis, cellular aging, and inflammation of UUO kidneys while IL1 β could eliminate these benefits. These data in the study could support that H3K9cr promoted cytokine-mediated signaling pathway by collaborating with other cells.

10. In figure 5D, there is a lack of correlation between the mRNA levels and the protein expression.

Response: Thanks for your careful reading. According to your suggestion, we have repeated the experiments and revised the data in Figure 5D.

11. Is there any correlation between ASCC2 expression / H3K9cr and kidney diseases?

Response: Thanks for your suggestion. In the revised manuscript, we have stained human kidney biopsy slides with H3K9cr and ACSS2 antibody (Fig S1G and Fig S9A). We found that both H3K9cr and ACSS2 were negatively correlated with kidney function indicator eGFR. However, with the limitation of sample size, they do not reach statistically significance.

12. The authors used the term “H3K9cr expression” it should be changed to “H3K9cr

levels”

Response: Thanks for your suggestion. According to your suggestion, we have corrected the term “H3K9cr expression” to “H3K9cr levels” in revised manuscript.

13. Some citations are missing. For examples, the paragraph that starts at line 50 is missing references.

Response: Thanks. According to your suggestion, we have checked all citations and added some important references. Unfortunately, as the journal guide suggested references should not exceed 70, we feel sorry that we have to quit some not those important references here. We are sorry to make you feel not comfortable as the limitation caused.

14. The text contains typos and grammatical errors that need to be revised.

Response: Thanks for your suggestions. With the help of ELSEVIER, we have revised the manuscript with Standard Language Editing.

Reviewer #3 (Remarks to the Author):

The authors conducted a functional study showing an increase of histone Kcr in fibrotic kidneys. Furthermore, the levels of Kcr (i.e. H3K9cr) were found to be reduced by higher levels of ACSS2 which in turn reduced IL1b expression and subsequently reduced macrophage activation and tubular cell senescence. These effects are finally attenuating kidney fibrosis. The paper is well written, the analyses are very comprehensive and sound, and the results generally support the conclusions.

1. My major concern is related to the causal link that the manipulation of histone Kcr by ACSS2 inhibition really results in a reduced kidney fibrosis state, which is needed to justify the claimed therapeutic potential. To underpin this relation, the authors presented several staining images (e.g., Fig. 3D, 2C, 7B) and blots (e.g. Fig. 1D, 1E, 2B, 3E, 5E, 7F). However, it is often difficult for me to see the differences between the fibrotic and reduced fibrotic tissue images, and the stated differences among groups of blots (taking into account that there is also variation in expression among controls). There should be a possibility to quantify these differences, or at least describe the specific changes in more detail (i.e., for the staining images).

Response: Thanks for your this useful suggestion. According to your comments, we have stained several typical kidney fibrotic makers such as COL6 and α -SMA in ACSS2^{-/-} and ACSS2 inhibitor-treated UUO mice. Furthermore, we have also quantified these staining to better understand the changes of fibrotic kidneys in our revision (Fig S7D-E, S26C and Fig 8C-D).

Additional comments are:

2. After reading the manuscript, it turned out that all experiments except the correlation of Histone Kcr expression with CKD was performed in mice or cell lines. This needs to be clearly stated in the manuscript, including the abstract, the methods and the figure legends.

Response: Thanks for your this useful suggestion. We have emphasized the data from mice and cell lines in abstract, methods and figure legends. Take figure legends for example, we have specified the sample species. “Fig. 1. Increased levels of crotonylation in **human** renal biopsies and kidneys from fibrotic **mice**”, “Fig.2. Global genetic knock out of ACSS2 in **mice** suppressed H3K9cr expression and alleviated fibrosis”, “Fig. 3. TEC-specific knock out of ACSS2 in **mice** suppressed H3K9cr expression and alleviated fibrosis”, “Fig. 4. Combination analysis of ChIP sequencing and RNA sequencing data in control and UUO of WT and ACSS2^{-/-} **mice**”. As the changes are a lot, please see our revision for more modifications.

3. Basis characteristics of the patients are missing. Although the sample size could be guessed from Fig. 1B, how many males/females were included, what was the age range, proportion of CKD? Which formula was used to estimate the eGFR? Are the correlation results in this figure comparable after adjusting for common covariates (e.g., sex and age)?

Response: Thanks a million for your suggestions. We have supplemented the basic characteristics (including sex, age, serum creatine, eGFR, and cause of diseases) of

these patients in Table S1 of our revision. The formula used to estimate the eGFR is Chronic Kidney Disease Epidemiology Collaboration equation (CKD-EPIcrea equation), and we also have added it in Table S1. In addition, we analyzed the data using multivariate regression to adjust sex and age influence. The results showed even adjusting sex and age, nuclear positive is still correlated with serum creatine and we supplemented these data in Table S2.

4. Almost all Figure descriptions contain many abbreviations, which are only explained in the main text. The Figure descriptions should be readable without having to go back to the main text to understand the abbreviations. Specifically, please describe the meaning of the abbreviations and of the arrows in Fig. 1A and Fig. S1A. Furthermore, I suggest to improve the description and/or labels of Fig. 1C and Fig. S9: only providing labels like WT_C1_cr is hard to follow.

Response: Thanks for your this useful suggestion. We are sorry to make you feel hard to read these figures, hoping you get better experience now. Corroding to your suggestion, we have added the meaning of abbreviations (such as IHC: immunohistochemical; pan-Kcr: pan anti-crotonyllysine; IF: Immunofluorescence; FAN: folic acid nephropathy; UUU: unilateral ureteric obstruction) of all figures in our revision.

As for the legend of Fig. 1A and Fig. S1A, we added the meaning of Triangle: “representative positive staining of pan-Kcr”. As to Fig.S9 which now has been Fig.S11, we have given more description in detail: “Fig. S11. ChIP-seq analysis in

each sample. (A) ChIP-seq analysis of test chromatin occupancy and heatmap of the distribution around ± 3 kb from the TSSs of H3K9ac in WT control mice. (B) ChIP-seq analysis of test chromatin occupancy and heatmap of the distribution around ± 3 kb from the TSSs of H3K9cr in WT control mice. (C) ChIP-seq analysis of test chromatin occupancy and heatmap of the distribution around ± 3 kb from the TSSs of input in WT control mice. (D) ChIP-seq analysis of test chromatin occupancy and heatmap of the distribution around ± 3 kb from the TSSs of H3K9ac in ACSS2^{-/-} control mice. (E) ChIP-seq analysis of test chromatin occupancy and heatmap of the distribution around ± 3 kb from the TSSs of H3K9cr in ACSS2^{-/-} control mice. (F) ChIP-seq analysis of test chromatin occupancy and heatmap of the distribution around ± 3 kb from the TSSs of input in ACSS2^{-/-} control mice. (G) ChIP-seq analysis of test chromatin occupancy and heatmap of the distribution around ± 3 kb from the TSSs of H3K9ac in WT UUO mice. (H) ChIP-seq analysis of test chromatin occupancy and heatmap of the distribution around ± 3 kb from the TSSs of H3K9cr in WT UUO mice. (I) ChIP-seq analysis of test chromatin occupancy and heatmap of the distribution around ± 3 kb from the TSSs of input in WT UUO mice. (J) ChIP-seq analysis of test chromatin occupancy and heatmap of the distribution around ± 3 kb from the TSSs of H3K9ac in ACSS2^{-/-} UUO mice. (K) ChIP-seq analysis of test chromatin occupancy and heatmap of the distribution around ± 3 kb from the TSSs of H3K9cr in ACSS2^{-/-} UUO mice. (L) ChIP-seq analysis of test chromatin occupancy and heatmap of the distribution around ± 3 kb from the TSSs of input in ACSS2^{-/-} UUO mice. UUO: unilateral ureteric obstruction; WT: wild type; ACSS2^{-/-}: ACSS2

knockout.” As the changes are a lot, please see our revision for more modifications.

5. Pathway analysis: it was stated that after ACSS2 deletion in mice the cytokine pathways decreased (Fig. S10 B-E). What does it mean exactly? As far as I see, the GeneRatio and p-values increased, and Count decreased which is the case for all pathways listed in these figure panels. Furthermore, please explain what GeneRatio and Count stand for and how to interpret them (also for Fig. 4E).

Response: Thanks for your question. Firstly, when we compared the RNA-seq data, there are several different genes between two groups. After analyzing these different genes, we could cluster these genes into different pathways. “Gene count” is the number of genes enriched in a GO (gene ontology) or KEGG (Kyoto Encyclopedia of Genes and Genomes) term. ‘Gene ratio’ is the percentage of total differential expression genes in the given GO/KEGG term. The “p-values” means whether these changes reach statistically significant. Taking Fig S12C for example, when we compare genes from WT UUO and WT Control, these increased genes could be clustered into several pathways, including cytokine-cytokine receptor interaction pathway. All of these pathways are increased pathway with statistically significant. When we compared ACSS2^{-/-} UUO with WT UUO (Fig S13A), cytokine-cytokine receptor interaction pathway is the top one changed pathway, which suggested that ACSS2 might regulate kidney fibrosis via this pathway. As for Fig 4E, it is similar as RNA-seq. The difference is that RNA-seq using RNA data, while ChIP-seq using ChIP data. Hope these explanations could help you have a better understanding.

6. Description of Fig. 4B: I assume it should state “tertile of gene expression” instead of “quartile”.

Response: Thank you so much for your careful reading! We have revised it into “tertile of gene expression”. Anyway, we have checked the manuscript and try our best to improve the writing.

7. Fig. S10 H: The figure description states qPCR analysis of H3K9Cr and H3K9ac ChIP products, but the labels show solely H3K9cr. Please check and clarify, which data is shown.

Response: Thank you again for your careful reading! We have corrected it in our new Fig 13E, which only showed the qPCR analysis of H3K9cr.

REVIEWER COMMENTS

Reviewer #1 (Remarks to the Author):

The author's thorough responses to my comments have significantly strengthened the manuscript, and I am pleased to recommend it for publication in Nature Communications.

Reviewer #2 (Remarks to the Author):

I have reviewed the revised manuscript, taking into account the changes made by the authors in response to our previous comments. While the authors have made some improvements in their manuscript, there is still a fundamental issue that remains unaddressed, which prevents me from recommending this paper for publication in Nature Communications. Specifically, our major concern is that the authors have not adequately addressed the potential function of other crotonylated residues, aside from H3K9cr, in the regulation of IL-1b expression. In light of the results presented in this manuscript, it would be misleading to conclude that IL-1b expression is solely regulated by the crotonylation levels of the H3K9 residue. To ensure the scientific rigor of this study, it is crucial that the authors conduct additional experiments and investigations to explore the potential roles of other crotonylated residues in the regulation of IL-1b expression. This is not just a minor point; it is a fundamental aspect of the study that should be addressed to provide a comprehensive understanding of the underlying mechanisms. I hope the authors will take these concerns into consideration and, if feasible, carry out the necessary experiments or provide a more comprehensive discussion of the existing data to strengthen their manuscript. Moreover, below are additional comments that were not satisfactorily addressed:

The following suggestions should strengthen your hypothesis:

Comment #4 – Figure S2: The authors should experimentally address this comment to test the effect of H3K14cr and H3K18cr (other ACSS2-regulated residues) on IL-1b expression.

Comment #7: The LC-MS/MS experiment is crucial to support the main conclusion of this manuscript. It is worth noting that this experiment can be conducted without the need for a Pan-Kcr antibody. Furthermore, there is a commercially available antibody for this purpose.

Reviewer #3 (Remarks to the Author):

I thank the authors for addressing my question in the revision. From my point of view, just a very few minor issues remain.

1. Thanks for providing details on the pathway analyses interpretation. I understand the pathway figures now, but the description in the manuscript is a misleading with respect to increasing/decreasing pathways. In detail, it should rather be written that the number of genes enriched in a pathway decreased (Figure S13A-B) (line 253). The sentence starting at line 260: "Enrichment plots from the GSEA results, the pathways 'cytokine-cytokine receptor interaction' and 'response to IL-1' were increased in UO kidneys while decreased after ACSS2 deletion (Fig. S13C)." is grammatically not correct. Furthermore, the enrichment score or the number of genes enriched in the pathway are increased/decreased (not the pathways itself). Finally, the description of "Gene count" and "Gene ratio" provided in the rebuttal letter should also be added to the corresponding Figure descriptions.

2. For Figure 4B, the "quartiles of gene expression" need to be changed to "tertiles" also in the main text.

3. The Figure S13(E) description in the supplement still states "(E) qPCR analysis of H3K9Cr and H3K9ac ChIP products from HEK-293T cells transfected with ACSS2 plasmids for 24 hours.", where "and H3K9ac" needs to be removed to my understanding.

Dear Reviewers,

We sincerely thank you for your comments on our manuscript entitled “Inhibition of ACSS2-mediated H3K9 crotonylation alleviates kidney fibrosis via IL-1 β -dependent macrophage activation and tubular cell senescence (No:NCOMMS-23-24253B)” In our 2nd revision, we have supplemented essential ChIP qPCR and western blot experiments with discussion, and corrected confused sentences to respond the reviewers’ comments. We have tried our best to revise and improve our manuscript according to your kind comments point to point. Please find the following detailed responses to your comments and suggestions.

Reviewer #1 (Remarks to the Author):

The author's thorough responses to my comments have significantly strengthened the manuscript, and I am pleased to recommend it for publication in Nature Communications.

Respond: Thanks for your comments, which is very helpful to our revision.

Reviewer #2 (Remarks to the Author):

I have reviewed the revised manuscript, taking into account the changes made by the authors in response to our previous comments. While the authors have made some improvements in their manuscript, there is still a fundamental issue that remains unaddressed, which prevents me from recommending this paper for publication in Nature Communications. Specifically, our major concern is that the authors have not adequately addressed the potential function of other crotonylated residues, aside from H3K9cr, in the regulation of IL-1b expression. In light of the results presented in this manuscript, it would be misleading to conclude that IL-1b expression is solely regulated by the crotonylation levels of the H3K9 residue. To ensure the scientific rigor of this study, it is crucial that the authors conduct additional experiments and investigations to explore the potential roles of other crotonylated residues in the regulation of IL-1b expression. This is not just a minor point; it is a fundamental aspect of the study that should be addressed to provide a comprehensive understanding of the underlying mechanisms. I hope the authors will take these concerns into consideration and, if feasible, carry out the necessary experiments or

provide a more comprehensive discussion of the existing data to strengthen their manuscript. Moreover, below are additional comments that were not satisfactorily addressed:

Respond: Thank you so much for your helpful suggestions. According to your comments, firstly, we have supplemented ChIP qPCR experiments to test the effect of H3K4cr, H3K14cr, H3K18cr, H3K23cr, H3K27cr, H3K36cr and H2BK34cr (all the antibody we could get) on IL-1b expression (Fig.S14C). The results were that the enrichment of IL-1b also could be increased by H3K14cr, H3K27cr H3K36cr and H2BK34cr under ACSS2-overexpressed tubular epithelial cells. In addition, according to your advice, we have added the limitation in our results and discussion as below.

1. Firstly, “However, whether crotonylation levels of other residue are also involved in regulating Il1b still needs further research.” (Marked in color under review mode)
2. Secondly, “Collectively, histone Kcr are involved in regulating renal disease, while besides H3K9, whether other residues of histone Kcr play significant roles still needs further experiments.” (Marked in color under review mode)
3. Thirdly, “We did not analyze in detail how this modification relates to known chromatin states (active versus poised versus heterochromatin), and do not explore the roles of other Kcrs in this paper. It is possible that crotonylation levels of other residue, not only H3K9, might be regulated by ACSS2 and could possibly play roles in regulating IL-1b-mediated renal fibrosis. Future LC-MS/MS analysis, ATAC sequencing and related experiments are needed to better understand the correlational and causal relationships between histone Kcr and overall chromatin remodeling, as

well as the correlation between different Kcrs during kidney fibrosis.” (Marked in color under review mode)

The following suggestions should strengthen your hypothesis:

Comment #4 – Figure S2: The authors should experimentally address this comment to test the effect of H3K14cr and H3K18cr (other ACSS2-regulated residues) on IL-1b expression.

Respond: Thank you again for your helpful suggestion. We fully agreed with your comments, and we do not summarize that IL-1b expression is solely regulated by the crotonylation levels of the H3K9 residue, where we mainly focused on the effects of H3K9 crotonylation in our previous submission. Here, we have also emphasized that the crotonylation levels of other residue might be regulated by ACSS2 and could possibly exert roles in renal fibrosis in this revision. According to your suggestion, we have performed the essential ChIP-qPCR experiments to test the effect of H3K4cr, H3K14cr, H3K18cr, H3K23cr, H3K27cr, H3K36cr and H2BK34cr (most of the commercial antibodies in market) on IL-1b expression. Interestingly, the enrichment of IL-1b could also be increased by H3K14cr, H3K27cr H3K36cr and H2BK34cr when we overexpress ACSS2, which we have added these results in Fig.S14C. And we emphasized “However, whether crotonylation levels of other residue are also involved in regulating Il1b still needs further research” in the last paragraph of “H3K9cr promoted cytokine production and regulated cytokine-cytokine receptor interaction in fibrotic kidneys” part and also in discussion. (Marked in color under

review mode)

Fig. S14. Combination analysis of ChIP sequencing/RNA sequencing data and the specificity of transcription factor binding motifs associated with H3K9cr.

Comment #7: The LC-MS/MS experiment is crucial to support the main conclusion of

this manuscript. It is worth noting that this experiment can be conducted without the need for a Pan-Kcr antibody. Furthermore, there is a commercially available antibody for this purpose.

Respond: Thank you again for your suggestion. We completely agree with your comments that the LC-MS/MS analysis are crucial, which is another key experiment to verify modification sites. According to your comment, we also try our best to finish ACSS2-dependent alterations in Kcr residues and find whether other residues behave like H3K9 in term of increase in Kcr but not Kac. In the revision, we have supplemented the changes of crotonylation and acetylation levels in other histone lysine of kidneys from ACSS2^{-/-} mice in Fig S7A -S7B. The knockout of ACSS2 could decrease the levels of H3K18cr and H3K27ac, while have no influence on H3K18ac and H3K27cr, which suggested the influence of ACSS2 on crotonylation and acetylation of histone lysine might be complicated. According to your suggestion, we further purchased these commercial Kcr and Kac antibody of other residues (H3K4, H3K14, H3K18, H3K23, H3K27 and H3K36) in market, and checked the modification changes of these residues in kidneys of ACSS2^{-/-} UUO mice, and TGFβ1-treated cell lines which were transfected by ACSS2 plasmids. We have supplemented the results Fig. S7 and Fig. S8.

Fig. S7. Global genetic knock out of ACSS2 influenced crotonylation and acetylation levels of several histone lysine residues in UUO-induced fibrotic kidneys.

Fig. S8. Overexpression of ACSS2 in tubular epithelial cells treated with TGFβ1 influenced crotonylation and acetylation levels of several histone lysine residues.

However, we did not perform the LC-MS/MS analysis, because we still need to do western blot analysis to confirm the LC-MS/MS results, where commercial antibodies of different site of histone might be not purchased or customized to product antibody.

At the same time, we also referenced to the publication and they researched the mechanism of enzyme-mediated-crotonylation without LC-MS/MS analysis to show the changes of some residues (Tang Xiaoqiang, et al. Circulation 2021). We are sorry that we cannot supply the experiments, however, in order to avoid misunderstand like you felt, we have discussed the possibility that ACSS2 might regulate other lysine as H3K9 in the same pattern and our limitations that failed to do LC-MS/MS analysis to check the changes of all residues in discussion part (Marked in color under review mode). We hope that we could gradually improve the role of crotonylation and acetylation modifications at various sites in renal fibrosis in the future. Thank you so much for your understanding.

Reviewer #3 (Remarks to the Author):

I thank the authors for addressing my question in the revision. From my point of view, just a very few minor issues remain.

1. Thanks for providing details on the pathway analyses interpretation. I understand the pathway figures now, but the description in the manuscript is a misleading with respect to increasing/decreasing pathways. In detail, it should rather be written that the number of genes enriched in a pathway decreased (Figure S13A-B) (line 253). The sentence starting at line 260: “Enrichment plots from the GSEA results, the pathways ‘cytokine-cytokine receptor interaction’ and ‘response to IL-1’ were increased in UUO kidneys while decreased after ACSS2 deletion (Fig. S13C).” is grammatically not correct. Furthermore, the enrichment score or the number of genes enriched in the pathway are increased/decreased (not the pathways itself). Finally, the description of “Gene count” and “Gene ratio” provided in the rebuttal letter should also be added to the corresponding Figure descriptions.

Respond: Thanks for your careful reading and helpful suggestions. We have revised “these two pathways decreased” into “the number of genes enriched in a pathway decreased”.

In addition, we have revised “Enrichment plots from the GSEA results, the pathways ‘cytokine-cytokine receptor interaction’ and ‘response to IL-1’ were increased in UUO kidneys while decreased after ACSS2 deletion.” into “According to the enrichment plots of the GSEA results, the number of genes enriched in the pathways “cytokine-cytokine receptor interaction” and “response to IL-1” were increased in

UUO kidneys while decreased after ACSS2 deletion ”.

Finally, the description of “Gene count” and “Gene ratio” has been added to Fig.4, and Fig.S13. In addition, we also checked the manuscript and tried our best to correct several confused sentences.

2. For Figure 4B, the “quartiles of gene expression” need to be changed to “tertiles” also in the main text.

Respond: Thanks for your careful reading. We have revised it into “tertiles”.

3. The Figure S13(E) description in the supplement still states “(E) qPCR analysis of H3K9Cr and H3K9ac ChIP products from HEK-293T cells transfected with ACSS2 plasmids for 24 hours.”, where “and H3K9ac” needs to be removed to my understanding.

Respond: Thanks for your careful reading again. We have removed “and H3K9ac”.

REVIEWERS' COMMENTS

Reviewer #2 (Remarks to the Author):

The authors have addressed my comments and strengthened the manuscript. Before publication in Nature Communications, two minor changes remain to be addressed:

1. The title of the manuscript should be changed according to the revised version of the manuscript since the authors did not establish the fact that kidney fibrosis is alleviated solely through H3K9 crotonylation. A better title would be : "Inhibition of ACSS2-mediated crotonylation alleviates kidney fibrosis via IL-1 β -dependent macrophage activation and tubular cell senescence".
2. Since ACSS2 regulates the crotonylation of all modified histone residues and as the authors show that other crotonylated histone residues might regulate IL-1 β production, I suggest to change the term "H3K9cr-mediated IL-1 β production".

Reviewer #3 (Remarks to the Author):

Thank you for addressing all my remaining issues. I have no further questions or concerns regarding the manuscript.

Dear editor and reviewers:

We sincerely thank you for your comments on our manuscript entitled “Inhibition of ACSS2-mediated H3K9 crotonylation alleviates kidney fibrosis via IL-1 β -dependent macrophage activation and tubular cell senescence (No: NCOMMS-23-24253C)”. In our 3rd revision, we have changed the title to “Inhibition of ACSS2-mediated crotonylation alleviates kidney fibrosis via IL-1 β -dependent macrophage activation and tubular cell senescence” according to the reviewer's suggestions. We earnestly appreciate your warm work, and we have tried our best to revise and improve our manuscript according to your kind comments. Please find the following detailed responses to your comments and suggestions.

Reviewer #2 (Remarks to the Author):

The authors have addressed my comments and strengthened the manuscript. Before publication in Nature Communications, two minor changes remain to be addressed: The title of the manuscript should be changed according to the revised version of the manuscript since the authors did not establish the fact that kidney fibrosis is alleviated solely through H3K9 crotonylation. A better title would be: “Inhibition of ACSS2-mediated crotonylation alleviates kidney fibrosis via IL-1 β -dependent macrophage activation and tubular cell senescence”. Since ACSS2 regulates the crotonylation of all modified histone residues and as the authors show that other crotonylated histone residues might regulate IL-1 β production, I suggest to change the term “H3K9cr-mediated IL-1 β production”.

Respond: Thanks for your comments. As suggested, we changed the title to “Inhibition of ACSS2-mediated crotonylation alleviates kidney fibrosis via IL-1 β -dependent macrophage activation and tubular cell senescence” (Marked in red under review mode).

Reviewer #3 (Remarks to the Author):

Thank you for addressing all my remaining issues. I have no further questions or concerns regarding the manuscript.

Respond: Thanks for your comments, which is very helpful to our revision.